# Exploring More to Solve More: Boosting Diversity in Text Diffusion Models via Entropy-Based Guidance

**Jingwei Zhang** [1]  **Haoyu Lei** [1]  **Zijin Feng** [2]  **Jiacheng Sun** [2]  **Farzan Farnia** [1]

## Abstract

Although diffusion models have revolutionized continuous domains like image synthesis through high quality generations and controllable guidance mechanisms, bringing this controllability to the discrete, sequential nature of text remains an open challenge. Meanwhile, current sampling strategies and guidance methods adjust token likelihoods without capturing the broader semantic landscape, leading to a suboptimal balance between fidelity and diversity. In this work, we introduce a novel training-free Semantic-Aware Kernel Entropy (SAKE) guidance method. Our method computes the order-2 Rényi entropy over a kernel Gram matrix that captures both cross-token semantic interactions and relative token positions. By linearizing this objective in the embedding space, we derive a tractable guidance signal that dynamically adjusts the sampling distribution—flattening it to encourage exploration during redundancy and sharpening it for fidelity when diverse. Empirical experiments demonstrate that our approach achieves a superior Pareto frontier between fidelity and diversity, and improves multi-sample performance on reasoning-intensive tasks, such as code and mathematics generation, compared to temperature scaling and discrete guidance baselines.

## 1. Introduction

Text generation has long been dominated by autoregressive (AR) models (Brown et al., 2020; Liu et al., 2024; Yang et al., 2025), which decompose the joint probability of a sequence into a product of conditional next-token probabilities. While highly effective, AR generation is inherently constrained by serial latency and a lack of bidirectional context during the decoding process. Recently, Diffusion Language Models (DLMs) have emerged as a compelling non-autoregressive alternative, offering a paradigm shift toward parallel decoding (Austin et al., 2021; Nie et al., 2025; Ye et al., 2025). The core mechanism of modern DLMs is a diffusion process operating directly within the discrete categorical token space. This involves a forward corruption process that gradually transitions valid tokens into noise, paired with a generative model trained to reverse this trajectory. Unlike autoregressive approaches, this framework enables simultaneous and iterative refinement of all tokens. Consequently, DLMs leverage global bidirectional context throughout generation and enable flexible trade-offs between computational cost and sample quality.

The success of continuous diffusion models in image and audio synthesis is largely driven by their controllability through inference-time guidance (Dhariwal & Nichol, 2021; Ho & Salimans, 2022; Bansal et al., 2023). Such guidance modifies the reverse diffusion dynamics to sample from desired conditional distributions without retraining. However, extending these paradigms to the discrete, sequential domain of text generation presents fundamental challenges. In continuous spaces, guidance usually operates by modifying the score function (the gradient of the data log-likelihood) via a differentiable energy function. In contrast, in the discrete domain, gradients are undefined, and the score function is based on a categorical posterior distribution. Modifying this distribution requires re-normalization, which is computationally intractable due to the need to calculate a partition function that sums over $V^L$ possible sequences, where $L$ is the sequence length and $V$ the vocabulary size. Although recent methods reduce this exponential cost to linear by assuming token independence (Sahoo et al., 2024), such factorizations ignore the complex cross-token correlations that are essential for coherent semantic formulation.

Besides computational intractability, discrete text diffusion suffers from a lack of semantic structure in the logit space, in contrast to the continuous diffusion that exploits the intrinsic semantic geometry of the continuous feature space.

[1]Department of Computer Science and Engineering, The Chinese University of Hong Kong [2]Huawei Foundation Model Department. Correspondence to: Jiacheng Sun <sunjiacheng1@huawei.com>, Farzan Farnia <farnia@cse.cuhk.edu.hk>.

*Proceedings of the 43rd International Conference on Machine Learning*, Seoul, South Korea. PMLR 306, 2026. Copyright 2026 by the author(s).

Because discrete models estimate transition probabilities over categorical distributions, the resulting logits encode model confidence rather than semantic similarity. Consequently, existing DLM sampling and guidance strategies (e.g., temperature scaling, D-CFG (Sahoo et al., 2024)) are limited to manipulating token likelihoods without awareness of the broader semantic landscape. To induce diversity, these methods rely on non-semantic distribution flattening via high-temperature scaling rather than navigating the textual manifold. This fundamentally limits the fidelity-diversity trade-off, as diversity is obtained through random noise injection into the probability mass function rather than through structured semantic exploration.

Diversity is often viewed as a trade-off against quality, but we argue that in complex reasoning, diversity in chain-of-thoughts (CoT) is a prerequisite for robustness rather than a mere source of variety. Prior work such as Self-Consistency (Wang et al., 2022) and Tree-of-Thoughts (Yao et al., 2023) shows that generating multiple distinct reasoning paths increases the probability of recovering the correct answer. Diversity prevents the model's probability distribution from collapsing onto a single, potentially erroneous chain of thought. When exploration is restricted to a narrow cluster of high-likelihood tokens, the model risks reinforcing its initial biases, leading to premature convergence on flawed reasoning. Consistent with this, our empirical results show that diversity-guided chains of thought significantly improves complex reasoning task pass rates, validating the utility of exploration in deterministic tasks.

In this paper, we study the guidance for DLMs in the discrete, sequential setting. Given a trained diffusion language model, we guide sampling to maximize task success in a way that maintains cross-token interactions, preserving diversity and semantic validity. Our contributions are threefold.

- We propose a general computationally efficient *diversity guidance* framework for discrete text diffusion. This framework overcomes the limitations of prior token-independent assumptions by integrating sequence-level semantic information. Building on this framework, we propose *Semantic-Aware Kernel Entropy* (SAKE), a tractable objective that models sequence semantic diversity, and explicitly maximize Rényi entropy of the sequence.

- We demonstrate that SAKE functions as an *adaptive distribution modulator*. Unlike static temperature scaling, our method dynamically adjusts the generation probability: it flattens the distribution to encourage exploration during mode collapse and sharpens it to ensure coherence when intrinsic diversity is already high.

- Empirical evaluations on both synthetic settings and standard LLM benchmarks confirm that our approach yields a superior trade-off between quality and diversity compared

to discrete guidance baselines and temperature-based sampling. Notably, applying SAKE to LLaDA-8B-base yields significant performance gains, improving HumanEval pass@32 ($41.1 \rightarrow 55.8$), MBPP pass@32 ($48.2 \rightarrow 56.1$), and GSM8K self-consistency ($71.5 \rightarrow 75.1$).

## 2. Related Works

### 2.1. Diffusion Models for Text Generation

Early text diffusion operated in continuous embedding spaces (Li et al., 2022; Yuan et al., 2022), but the performance is limited compared to AR baselines due to a challenge mapping denoised continuous vector back to a coherent discrete token. To address the embedding bottleneck, recent research has shifted toward discrete diffusion processes that operate directly on the categorical token space. Foundational work D3PM (Austin et al., 2021) formalized diffusion over discrete states using transition matrices, defining noise processes such as uniform corruption or corruption to a specific mask token (absorbing state). Building on this, modern Diffusion Language Models (DLMs) have scaled significantly (Lou et al., 2023; Sahoo et al., 2024; Ou et al., 2024; Nie et al., 2025; Ye et al., 2025), and demonstrated that DLMs can match or exceed AR models in perplexity and zero-shot generation while enabling arbitrary infilling and iterative refinement.

### 2.2. Guidance Mechanisms for Diffusion Models

Continuous score-based diffusion models (Sohl-Dickstein et al., 2015; Ho et al., 2020; Song et al., 2020; Song & Ermon, 2019) has shown remarkable abilities in various generation tasks, including images (Rombach et al., 2022; Dhariwal & Nichol, 2021; Ho et al., 2022) and videos (He et al., 2022; Blattmann et al., 2023a;b). Conditional generation is achieved by guidance, modifying the learned score function during inference (Dhariwal & Nichol, 2021; Nichol et al., 2021; Bansal et al., 2023; Ho & Salimans, 2022; Corso et al., 2024; Askari Hemmat et al., 2024; Jalali et al., 2025a; Sani et al., 2026; Jalali et al., 2026). As mentioned earlier, transferring these techniques to discrete diffusion is mathematically non-trivial. To make guidance tractable, Schiff et al. (2024) proposed Discrete Classifier-Free Guidance (D-CFG) and Discrete Classifier-Based Guidance (D-CBG), which rely on an independence assumption that assumes token transitions are independent. Although this reduces the normalization cost from exponential to linear with sequence length, the independence assumption ignores the cross-token correlations that are vital for semantic formulation.

### 2.3. Diversity in Generation Tasks

Beyond generation quality, diversity has increasingly become a critical value within the research community. Var-

ious metrics have been proposed to quantify diversity in both image (Sajjadi et al., 2018; Jalali et al., 2023; Ospanov et al., 2024; Ospanov & Farnia, 2025; Ospanov et al., 2025) and text (Zhu et al., 2025) domains. In auto-regressive text generation, several methods extending beyond temperature scaling have been developed to enhance output variety (Vijayakumar et al., 2016; Holtzman et al., 2019; Nguyen et al., 2024). Also, the entropy-based novelty of sample generation has been studied in (Zhang et al., 2024; 2025; Lotfian et al., 2026), and the related works (Hu et al., 2025a; Rezaei et al., 2025; Hu et al., 2025b; Jafari & Farnia, 2026; Nia & Farnia, 2026) study online diversity-aware evaluation of generative models. The role and comparison of embeddings for diversity evaluation has been analyzed in (Stein et al., 2023; Jalali et al., 2025b; Wu et al., 2025; Gong et al., 2025; Wu & Farnia, 2026; Wu et al., 2026). Meanwhile, diffusion models have demonstrated remarkable capability in controlling diversity for image generation, due to the flexible inference mechanisms. However, while training-free guidance techniques have proven effective for continuous image diffusion (Sadat et al., 2024; Corso et al., 2024; Kirchhof et al., 2025; Jalali et al., 2025a), the diversity guidance in discrete diffusion models for text generation is unexplored.

## 3. Preliminaries

**Continuous Diffusion Generative Models.** Diffusion Models (DMs) (Ho et al., 2020; Song & Ermon, 2019; Song et al., 2020) are generative models designed to sample from a target data distribution $p_{\text{data}}(\mathbf{x}_0)$ by reversing a predefined forward noising process (Sohl-Dickstein et al., 2015). In the forward diffusion process, a data sample $\mathbf{x}_0$ is progressively perturbed with Gaussian noise over a continuous time interval $t \in [0, T]$. This process is mathematically described as adding noise to obtain a noisy state $\mathbf{x}_t = \sqrt{\alpha_t}\mathbf{x}_0 + \sqrt{1 - \alpha_t}\boldsymbol{\epsilon}_t$, where $\boldsymbol{\epsilon}_t \sim \mathcal{N}(\mathbf{0}, \mathbf{I})$ represents standard Gaussian noise, and $\alpha_t \in [0, 1]$ is a monotonically decreasing schedule controlling the noise level. DMs (Ho et al., 2020) train a neural network $\boldsymbol{\epsilon}_\theta : \mathcal{X} \times [T] \mapsto \mathcal{X}$ to predict the noise $\boldsymbol{\epsilon}_t$ at each time step $t$. This objective implicitly learns the *score function* of the marginal distribution $p_t(\mathbf{x}_t)$ (Song & Ermon, 2019; Song et al., 2020):

$$\min_\theta \mathbb{E}_{\mathbf{x}_t, \boldsymbol{\epsilon}_t, t} \left[ \left\| \boldsymbol{\epsilon}_\theta(\mathbf{x}_t, t) + \sqrt{1 - \alpha_t} \underbrace{\nabla_{\mathbf{x}_t} \log p_t(\mathbf{x}_t)}_{\text{Score Function}} \right\|_2^2 \right],$$
(1)

During inference, samples are generated by solving the reverse-time SDE from $t = T$ to $t = 0$. Crucially, there exists a corresponding deterministic process known as the *probability flow ODE* (PF-ODE), whose trajectories share the same marginal distributions $\{p_t(\mathbf{x}_t)\}_{t \in [0, T]}$ as the SDE (Song et al., 2020; Lipman et al., 2022).

For conditional generation tasks like text-to-image synthesis, the objective of diffusion models is to learn the conditional distribution $p(\mathbf{x}_0 \mid y)$ given a condition $y$. Following the score matching framework (Song & Ermon, 2019; Song et al., 2020), the corresponding conditional score can be expressed as:

$$\underbrace{\nabla_{\mathbf{x}_t} \log p_t(\mathbf{x}_t | y)}_{\text{Conditional Score}} = \underbrace{\nabla_{\mathbf{x}_t} \log p_t(\mathbf{x}_t)}_{\text{Unconditional Score}} + \underbrace{\nabla_{\mathbf{x}_t} \log p_t(y | \mathbf{x}_t)}_{\text{Guidance}}.$$
(2)

**Discrete Diffusion Process.** The forward diffusion process progressively corrupts discrete data, typically toward an absorbing state (e.g., a mask token) or a high-entropy (often uniform) noise distribution over the vocabulary (Austin et al., 2021). In principle, this process could be modeled as a time-inhomogeneous continuous-time Markov chain on a finite state space $\mathcal{X}$. The process is governed by a rate matrix $Q_t \in \mathbb{R}^{|\mathcal{X}| \times |\mathcal{X}|}$ for each time $t$, where the off-diagonal entries are non-negative and columns sum to zero. The evolution of the probability mass function (PMF) $p_t \in \mathbb{R}^{|\mathcal{X}|}$ over the states is described by the Kolmogorov forward equation:

$$\frac{d}{dt} p_t = Q_t p_t.$$
(3)

The corresponding reverse process, which generates data by reversing the corruption, is also a Markov process. Its exact transition rates $\overline{Q}_t$ are usually defined using the forward rates and the *concrete score* ratio (Meng et al., 2022), $r_t(y \mid x) = p_t(y)/p_t(x)$:

$$\overline{Q}_t(y, x) = Q_t(x, y) r_t(y \mid x),$$
(4)

with the diagonal entries defined as $\overline{Q}_t(x, x) = -\sum_{y \neq x} \overline{Q}_t(y, x)$. As the true score ratio $r_t$ is intractable, in practice it is approximated with a neural network $s_\theta(x, t)[y] \approx r_t(y \mid x)$. This yields a learned, normalized reverse transition probability kernel for $y \neq x$:

$$P_t(y \mid x) = \frac{Q_t(x, y) s_\theta(x, t)[y]}{\sum_{z \neq x} Q_t(x, z) s_\theta(x, t)[z]}.$$
(5)

## 4. Methodology

### 4.1. Unified Framework for Text Diffusion Guidance

Let $\mathcal{V}$ denote the vocabulary, and the base model's logits denoted as $\mathbf{z} \in \mathbb{R}^{|\mathcal{V}|}$, inducing a probability distribution $P_{\text{base}}(y) = \text{softmax}(\mathbf{z})_y$ over tokens $y \in \mathcal{V}$.

We define a generalized class of guided distribution $P_\gamma$, parameterized by a guidance signal vector $\psi \in \mathbb{R}^{|\mathcal{V}|}$ and a guidance scalar $\gamma \in \mathbb{R}$, as

$$P_\gamma(y) = \frac{1}{Z(\gamma)} P_{\text{base}}(y) \cdot \exp\left(\gamma \cdot \psi_y\right),$$
(6)

where $Z(\gamma)$ is the partition function. This formulation encapsulates a broad class of sampling strategies. To characterize the behavior of these strategies, we analyze the local sensitivity of the Shannon entropy $H[P_\gamma]$ with respect to the guidance scalar $\gamma \in \mathbb{R}$.

**Proposition 4.1** (Entropy Dynamics of Text Diffusion Guidance). *The first-order change in the Shannon entropy of the guided distribution $P_\gamma$ at the limit of zero guidance is governed by the negative covariance between the base log-probabilities and the guidance signal:*

$$\nabla_\gamma H[P_\gamma]|_{\gamma=0} = -\mathrm{Cov}_{Y \sim P_{\text{base}}}\big[\log P_{\text{base}}(Y), \psi_Y\big]. \quad (7)$$

*Since $\log P_{\text{base}}(y) = \mathbf{z}_y - \log Z(0)$, and covariance is translation invariant under constant shifts (such as $\log Z(0)$), the gradient simplifies to:*

$$\nabla_\gamma H[P_\gamma]|_{\gamma=0} = -\mathrm{Cov}_{P_{\text{base}}}[\mathbf{z}, \psi]. \quad (8)$$

*Proof.* We defer the proof to the Appendix $\qquad\square$

This proposition establishes that guidance acts as an entropy reducer if and only if the signal $\psi$ is positively correlated with the base model's logits $\mathbf{z}$.

### 4.2. Analysis of Existing Training-free Guidance Mechanisms

We now apply Proposition 4.1 to characterize the behavior of existing strategies.

**Temperature Scaling.** Temperature scaling is a commonly used sampling strategy that rescales the base logits by a temperature parameter $\tau$. For $0 < \tau < 1$, standard temperature sampling is equivalent to setting the scalar strength $\gamma = \frac{1}{\tau} - 1$ and defining the guidance signal $\psi$ for temperature scaling as the logits themselves, i.e., $\psi_{\text{temp}} = \mathbf{z}$. Substituting this into Proposition 4.1 yields:

$$\nabla_\gamma H|_{\gamma=0} = -\mathrm{Var}_{P_{\text{base}}}[\mathbf{z}] \leq 0. \quad (9)$$

Since variance is strictly non-negative, temperature scaling monotonically reduces entropy. More broadly, alternative sampling strategies such as Top-$k$ and Nucleus sampling explicitly truncate the probability distribution, which inevitably constrains the diversity of generated outputs.

*Remark* 4.2. The behavior could be reversed by setting $\tau > 1$ which leads to a negative $\gamma$. In this case, the guidance move in the opposite direction and monotonically increases entropy. This behavior inevitably increases the likelihood of all noisy tokens. We empirically demonstrate in Section 5 that high temperature increases diversity at a significant cost of generation quality.

**Discrete Classifier-Free Guidance (D-CFG).** D-CFG extrapolates between an unconditional estimate $\mathbf{z}_\emptyset$ and a conditional estimate $\mathbf{z}_c$. Treating the unconditional distribution as the base $P_{\text{base}}$, the guidance signal is given by the residual vector $\psi_{\text{cfg}} = \mathbf{z}_c - \mathbf{z}_\emptyset$. Substituting this into Proposition 4.1 and exploiting the linearity of covariance, the entropy gradient decomposes into two competing terms:

$$\nabla_\gamma H|_{\gamma=0} = \mathrm{Var}_{P_\emptyset}[\mathbf{z}_\emptyset] - \mathrm{Cov}_{P_\emptyset}[\mathbf{z}_\emptyset, \mathbf{z}_c]. \quad (10)$$

D-CFG typically acts as an entropy reducer by sharpening the distribution around the mode, it increases entropy only in specific cases of misalignment between the conditional and prior models (characterized by low covariance between $\mathbf{z}_\emptyset$ and $\mathbf{z}_c$).

In summary, existing control mechanisms are less suited for enhancing meaningful diversity. Although temperature scaling can increase entropy, it does so at the cost of injecting indiscriminate noise, whereas truncation methods are explicitly designed to suppress the distribution tail. Similarly, D-CFG lacks a consistent mechanism to promote variation. Furthermore, these standard methods operate exclusively in the logit space, which primarily reflects model confidence rather than semantic content.

The sharpening behavior of standard methods, combined with the lack of semantic awareness in the logit space, fundamentally limits diversity in text generation tasks. To overcome these limitations, we propose SAKE guidance in the following subsection, a mechanism that leverages the semantic information inherent in the sequence to actively promote diverse generation. SAKE also supports bi-directional entropy modulation, which allows the model to dynamically switch between exploration and exploitation.

### 4.3. Our Method: Semantic-Aware Kernel Entropy Guidance

In this subsection, we proceed in three steps. First, we formalize the general discrete diversity signal used for guidance. Second, we resolve the intractability of the discrete formulation via a computationally efficient embedding-space linearization. Finally, we introduce a semantic-aware diversity function that improve sequence spectral diversity by maximizing the order-2 Rényi entropy.

#### 4.3.1. DISCRETE DIVERSITY GUIDANCE

**General Discrete Diversity Guidance Signal.** We first define the general discrete diversity signal for DLMs as:

$$\psi_{\text{div}}(y) = \mathcal{D}\big(y; \mathbf{x}_{-i}\big) - \mathcal{D}\big(x_i; \mathbf{x}_{-i}\big) \quad (11)$$

where $\mathcal{D} : \mathbb{N}^L \to \mathbb{R}$ is the discrete diversity function, $L$ is the sequence length, $i \leq L$ is the token index, $\mathbf{x}_{-i}$ is the sequence except token at index $i$. In this case, $\psi_{\text{div}}(y)$ measures the diversity "gap" (or potential difference) between a proposal for the next state $y \in \mathcal{V}$ and the current state $x_i$. Therefore, a token that brings more diversity gain will be

**Algorithm 1** Diversity Guidance for Discrete Text Diffusion

---

**Require:** Pretrained DLM $f_\theta$, Prompt $\mathbf{c}$, Number of steps $T$, Guidance scale $\gamma$, Bandwidths $\sigma, \sigma_{\text{attn}}$, Token embedding matrix $E \in \mathbb{R}^{|\mathcal{V}| \times d}$, masking schedule $N(t)$.

1: **Initialize:** $\mathbf{x}_0 = [\mathbf{c}, [\text{M}], \dots, [\text{M}]]$
2: **for** $t = 1$ to $T$ **do**
3:      Logits $\mathbf{Z}_t$, Embedding $\mathbf{H}_t = f_\theta(\mathbf{x}_{t-1})$
4:      Compute semantic kernel $\mathbf{K}$ where $\mathbf{K}_{ij} = \exp(-\|\mathbf{h}_i - \mathbf{h}_j\|^2/2\sigma^2)$
5:      Compute attention kernel $\mathbf{K}_{\text{attn}}$ where $\mathbf{K}_{\text{attn},ij} = \exp(-\|i - j\|^2/2\sigma_{\text{attn}}^2)$
6:      Initialize $G = \mathbf{0}^{d \times L}$
7:      **for** $i = 1$ to $L$ **do**
8:          $\mathbf{g}_i = \sum_{j \neq i}(\mathbf{K}_{\text{attn},ij} \cdot \mathbf{K}_{ij})^2 \cdot (\mathbf{h}_i - \mathbf{h}_j)$
9:          $G[:, i] = \frac{2}{\sigma^2}\mathbf{g}_i$
10:      **end for**
11:      Guidance Signal $\mathbf{\Psi}_{\text{div}} = E\,G$
12:      $\tilde{\mathbf{Z}}_t = \mathbf{Z}_t + \gamma \cdot \mathbf{\Psi}_{\text{div}}$
13:      Sample $\mathbf{x}_t$ based on $N(t)$ and guided logits $\tilde{\mathbf{Z}}_t$
14: **end for**

---

more likely generated after guidance. In combination with Proposition 4.1, the sign of the covariance now becomes indefinite. Unlike static sharpening methods, diversity guidance creates a dynamic feedback loop:

- **Exploration (entropy increase):** If the base prediction $\mathbf{z}^{(i)}$ is semantically redundant (highly similar to other elements $\mathbf{z}^{(j)}$), the signal $\psi_{\text{div}}^{(i)}$ will oppose the direction of $\mathbf{z}^{(i)}$. This yields a negative covariance, increasing entropy and encouraging the model to explore.

- **Exploitation (entropy decrease):** If the base prediction is already diverse, the difference term vanishes, allowing the natural confidence of the model to dominate.

However, considering the size of $\psi_{\text{div}}$ is $|\mathcal{V}|$, and current DLMs usually adopt a huge vocabulary size in practice. Guiding a sequence of length $L$ will lead to $L \times |\mathcal{V}|$ evaluations of the diversity function, which is unacceptable for real-time DLM inferences. We will first show how to efficiently calculate the diversity signal.

### 4.3.2. EFFICIENT COMPUTE VIA EMBEDDING-SPACE LINEARIZATION.

To avoid the brute-force evaluation of the discrete diversity gap, we linearize the diversity function within the continuous embedding space. This allows us to replace the discrete difference with a gradient projection.

Let $\mathbf{h}_y$ and $\mathbf{h}_{x_i}$ denote the continuous embedding vectors for a candidate token $y$ and the current token $x_i$, respectively. Let $\mathbf{H}_{x_{-i}}$ be the matrix of token embeddings for

a sequence except token $i$. Assuming continous diversity function $\tilde{\mathcal{D}}$ is differentiable with respect to the embedding of the $i$-th token, the diversity potential for candidate $y$ can be approximated as:

$$\tilde{\psi}_{\text{div}}^{(i)}(y) \triangleq \tilde{\mathcal{D}}(\mathbf{h}_y; \mathbf{H}_{x_{-i}}) - \tilde{\mathcal{D}}(\mathbf{h}_{x_i}; \mathbf{H}_{x_{-i}}) \quad (12)$$

$$\approx \left\langle \nabla_{\mathbf{h}_{x_i}} \tilde{\mathcal{D}}(\mathbf{h}_{x_i}; \mathbf{H}_{x_{-i}}), \mathbf{h}_y - \mathbf{h}_{x_i} \right\rangle, \quad (13)$$

since $\mathbf{h}_{x_i}^T \nabla_{\mathbf{h}_{x_i}} \tilde{\mathcal{D}}(\mathbf{h}_{x_i}; \mathbf{H}_{x_{-i}})$ is a fixed term for all $y$ and softmax is shift-invariant, we may ignore this term and the final guidance signal can be approximated as:

$$\tilde{\psi}_{\text{div}}^{(i)} \approx E\,\nabla_{\mathbf{h}_{x_i}} \tilde{\mathcal{D}}(\mathbf{h}_{x_i}; \mathbf{H}_{x_{-i}}), \quad (14)$$

where $E \in \mathbb{R}^{|\mathcal{V}| \times d}$ is the vocabulary embedding matrix containing all candidate tokens. This approximation reduces the computation from $L \times |\mathcal{V}|$ evaluations of discrete diversity function $\mathcal{D}$ to a single backward pass of a continuous diversity function $\tilde{\mathcal{D}}$. This formulation establishes a generic, computationally efficient framework for diversity guidance, applicable to any differentiable diversity metric $\tilde{\mathcal{D}}$ without incurring the latency costs of brute-force search.

Eq. 13 relies on a first-order Taylor expansion, assuming a smooth continuous diversity function $\tilde{D}$ and bounded embedding distances $\|\mathbf{h}_\mathbf{y} - \mathbf{h}_{\mathbf{x}_\mathbf{i}}\|$. These assumptions hold in modern LLMs (like LLaDA) because LayerNorm/RMSNorm constrains embeddings, and Gaussian kernel in our $\tilde{D}$ is smooth.

A theoretical limitation of this linear approximation is that, unlike the true bounded RBF kernel, it does not plateau. Under large guidance scales ($\gamma$), an outlier token lying far along the gradient could receive an overestimated diversity score. Fortunately, this edge case is controllable. By restricting the guidance signal to a 'trust region', simply applying a mild Top-p or Top-k filter to the base logits before guidance, we ensure diversification only occurs among tokens that are semantically plausible.

### 4.3.3. SAKE: SEMANTIC-AWARE KERNEL ENTROPY GUIDANCE

Having established the efficient diversity guidance framework, we now instantiate the continuous objective $\tilde{\mathcal{D}}$ to specifically target semantic redundancy. We propose maximizing the Rényi entropy of the kernel Gram matrix to promote *sequence* spectral diversity grounded in information-theoretic principles.

We used two types of kernel. One is the *semantic* kernel, which measures the semantic similarity between tokens. The other one is the *attention* kernels, which measures the weights that tokens attend to each other in sequences. In our implementation, we leverage the relative positional information of tokens to determine attention weights.

**Proposition 4.3** (SAKE Guidance Signal). *Let* $\mathbf{H} = [\mathbf{h}_1, \ldots, \mathbf{h}_L] \in \mathbb{R}^{d \times L}$ *be the matrix of token embeddings for a sequence of length $L$. Consider the Gaussian RBF kernel $\kappa(\mathbf{h}_i, \mathbf{h}_j) = \exp(-\|\mathbf{h}_i - \mathbf{h}_j\|^2/2\sigma^2)$ with bandwidth $\sigma > 0$, and let $\mathbf{K} \in \mathbb{R}^{L \times L}$ be the kernel Gram matrix with $\mathbf{K}_{ij} = \kappa(\mathbf{h}_i, \mathbf{h}_j)$. Similarly, $\mathbf{K}_{attn,(i,j)} = \exp(-\|i-j\|^2/2\sigma_{attn}^2)$. The normalized Hadamard product of kernel Gram matrices serves as a valid quantum density matrix. The order-2 Rényi entropy of this representation is:*

$$H_2\left(\frac{1}{L}\mathbf{K} \odot \mathbf{K}_{attn}\right) = -\log\left(\frac{1}{L^2}\|\mathbf{K} \odot \mathbf{K}_{attn}\|_F^2\right). \quad (15)$$

*To increase $H_2$ by adjusting the $i$-th token embedding $\mathbf{h}_i$, we define the continuous diversity function:*

$$\tilde{\mathcal{D}}_\kappa(\mathbf{h}_i; \mathbf{H}_{-i}) \triangleq -\sum_{j \neq i} \kappa_{attn}(i,j)^2 \cdot \kappa(\mathbf{h}_i, \mathbf{h}_j)^2. \quad (16)$$

*where $\mathbf{H}_{-i}$ denotes the embeddings of all tokens except the $i$-th. The gradient of this function with respect to $\mathbf{h}_i$ is:*

$$\nabla_{\mathbf{h}_i}\tilde{\mathcal{D}}_\kappa = \frac{2}{\sigma^2}\sum_{j \neq i} \kappa_{attn}(i,j)^2 \cdot \kappa(\mathbf{h}_i, \mathbf{h}_j)^2(\mathbf{h}_i - \mathbf{h}_j). \quad (17)$$

*Proof.* We defer the proof to the Appendix □

The gradient $\nabla_{\mathbf{h}_i}\tilde{\mathcal{D}}_\kappa$ has an intuitive geometric interpretation as a weighted sum of repulsive forces, where each contextual token $\mathbf{h}_j$ pushes $\mathbf{h}_i$ away with strength proportional to $\kappa_{attn}(i,j)^2\kappa(\mathbf{h}_i, \mathbf{h}_j)^2$. By substituting $\nabla_{\mathbf{h}_i}\tilde{\mathcal{D}}_\kappa$ into the first-order approximation from Eq. 14, we obtain the computationally efficient SAKE guidance signal:

$$\tilde{\psi}_{\mathrm{div}}^{(i)} \approx E\nabla_{\mathbf{h}_i}\tilde{\mathcal{D}}_\kappa = \frac{2E}{\sigma^2}\sum_{j \neq i}\kappa_{attn}(i,j)^2 \cdot \kappa(\mathbf{h}_i, \mathbf{h}_j)^2(\mathbf{h}_i - \mathbf{h}_j), \quad (18)$$

where $E \in \mathbb{R}^{|\mathcal{V}| \times d}$ is the vocabulary embedding matrix. This formulation directly connects kernel-based order-2 Rényi entropy maximization to practical sequence generation with efficient spectral diversity guidance, which has linear sequence length complexity.

Algorithm 1 shows the pseudocode of our proposed SAKE method. The algorithm takes as input a pretrained DLM $f_\theta$, a prompt $\mathbf{c}$, the number of diffusion steps $T$, guidance scale $\gamma$, kernel bandwidths, an embedding matrix $E$, and a masking schedule. Line 1 first initializes a sequence with the prompt followed by masked tokens. At each step $t$, the model produces logits and token embeddings in line 3, from which semantic and attention kernels are computed to capture token similarity and positional relevance in lines 4-5. Lines 7–11 then compute the gradient of the diversity function, which is projected onto the vocabulary to form the diversity guidance signal in line 12. In line 13, the signal

is added to the base logits to obtain entropy-modulated logits, which are then used to sample the next sequence state according to the diffusion schedule in line 14. After $T$ steps, the procedure terminates and outputs the final sequence $x_T$.

# 5. Numerical Results

## 5.1. Gaussian Synthesis

To compare temperature scaling with our diversity guidance, we use a 2D Gaussian mixture model for precise evaluation (Figure 4). The baseline (top) shows that increasing temperature $T$ improves mode coverage but significantly sacrifices fidelity, resulting in diffuse samples. In contrast, our method (bottom) with varying strength $\gamma$ achieves broad mode coverage while maintaining tight alignment with the target components. This confirms that our approach effectively prevents mode collapse without the quality degradation inherent to high-temperature sampling. The detailed analysis is provided in Appendix B.

## 5.2. Diversity-Quality Pareto Frontier Experiments

We analyze the Diversity-Quality Pareto frontier to evaluate the trade-off between generation quality and variety. Figure 2 demonstrates that *diversity-guided* sampling consistently yields superior frontiers compared to standard *temperature sampling*. While temperature scaling forces a strict trade-off—sacrificing quality for diversity—our method shifts the frontier outward, maintaining higher quality at comparable diversity levels across all tasks.

**Gaussian Mixture Generation (Fig. 2, top-left)**: Using Coverage (diversity) and Density (quality) metrics (Naeem et al., 2020), we find that increasing temperature $T$ expands coverage but causes a precipitous drop in density. Conversely, sweeping guidance strength $\gamma$ shifts the frontier outward, achieving higher coverage for equivalent density levels compared to the baseline.

**Arithmetic Series Generation (Fig. 2, top-right)**: In this controlled setting, we measure *Validity* (correctness) against *Uniqueness*. High-temperature sampling improves uniqueness only at the cost of a sharp drop in validity. Diversity guidance significantly extends the Pareto frontier, enabling high uniqueness with minimal loss in validity.

**Story Continuation (Fig. 2, bottom-left)**: We analyze the trade-off between Perplexity (quality) and Distinct-2 (diversity) (Li et al., 2016) using LLaDA (Nie et al., 2025). Results indicate that aggressive temperature sampling incurs steep perplexity penalties for marginal diversity gains. Diversity guidance achieves a more favorable balance, outperforming baselines (including CFG) particularly in high-diversity regimes.

**Brainstorm Generation (Fig.2, bottom-right)**: This task,

## Gaussian Mixture Synthesis for Discrete Diffusion Reverse Process

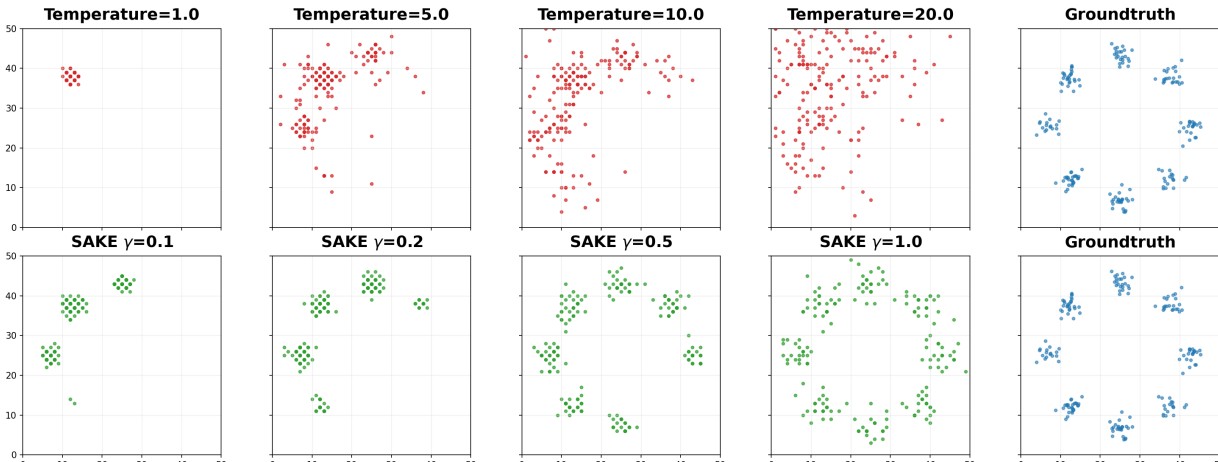

*Figure 1.* Comparison of sampling behaviors for a 2D 8-Gaussian-mixture target. **Top row**: baseline unguided reverse process at different temperatures $T \in \{1, 5, 10, 20\}$, illustrating a coverage-fidelity trade-off (higher $T$ covers more modes but yields more diffuse, lower-quality samples). **Bottom row**: Our SAKE with guidance strengths $\gamma \in \{0.1, 0.2, 0.5, 1.0\}$, which improves mode coverage while maintaining better precision. Rightmost column shows ground-truth samples from the 8-Gaussian mixture.

requiring multiple creative proposals per prompt, presents a challenge for baselines. Temperature sampling suffers rapid quality degradation at high diversity, while D-CFG remains overly conservative. In contrast, diversity guidance establishes a superior frontier, maintaining low perplexity even as Distinct-2 increases. Qualitative examples (Table 1) further confirm that while baselines collapse into repetitive syntactic templates, SAKE successfully diversifies semantic content without sacrificing coherence.

Collectively, these results suggest that SAKE is a more effective strategy for traversing the quality-diversity manifold than relying solely on stochastic temperature scaling. The detailed analysis is provided in Appendix C.

### 5.3. Diversity-Guided Chain-of-Thought in Complex Reasoning Tasks

Table 2 summarizes performance on HumanEval, MBPP, and GSM8K using LLaDA-8B. We compare the unguided base model, D-CFG, and our SAKE using standard stochastic sampling. We report Pass@32 ($n{=}32$) for code tasks and self-consistency ($n{=}5$) for GSM8K. To ensure a fair comparison, we perform a hyperparameter search for both the baseline methods and ours via grid search.

Our method boosts performance in settings requiring diverse exploration. On HumanEval, while Pass@1 remains competitive, Pass@32 improves substantially to 55.8, outperforming both the base model (41.1) and D-CFG (41.2). Similarly, on MBPP, Pass@32 rises to 56.1 (vs. base 48.2). This indicates our method generates higher *effective* diversity, producing a candidate set with a higher probability of correctness rather than merely increasing surface variability.

Figure 3 highlights the mitigation of mode collapse. At low temperature ($T{=}0.2$), the baseline saturates quickly ($0.33 \rightarrow 0.41$), whereas our method forces exploration, improving Pass@32 to 0.56. Notably, our method at $T{=}0.2$ matches the baseline's performance at $T{=}0.7$, demonstrating the ability to extract latent knowledge without the quality degradation risks associated with high-temperature sampling. For GSM8K (5-shot), our method achieves 75.1 (strict) and 73.5 (flexible), surpassing both the base model (71.5/68.8) and D-CFG. These results confirm that diversity guidance benefits downstream selection mechanisms (Pass@$k$, majority voting) by reallocating probability mass toward distinct, plausible reasoning paths.

### 5.4. Computation Efficiency

As shown in Eq.18, the SAKE diversity signal possesses linear sequence length complexity and can be efficiently calculated. We empirically validate this theoretical efficiency in Table 3, which reports the generation speed (tokens/sec, batch size equals 1) across varying prompt lengths for the LLaDA model in a single GPU.

The results demonstrate that SAKE introduces negligible inference latency. On average, our method maintains a generation speed of 44.09 tokens/sec, retaining approximately 93% of the unguided baseline's throughput (47.41 tokens/sec). In contrast, D-CFG suffers from a substantial computational penalty, averaging only 31.83 tokens/sec—a reduction of nearly 33% compared to the Baseline.

This efficiency gap becomes more significant at longer context windows. At a prompt length of 1024, SAKE remains highly efficient (21.51 tokens/sec) compared to the Base-

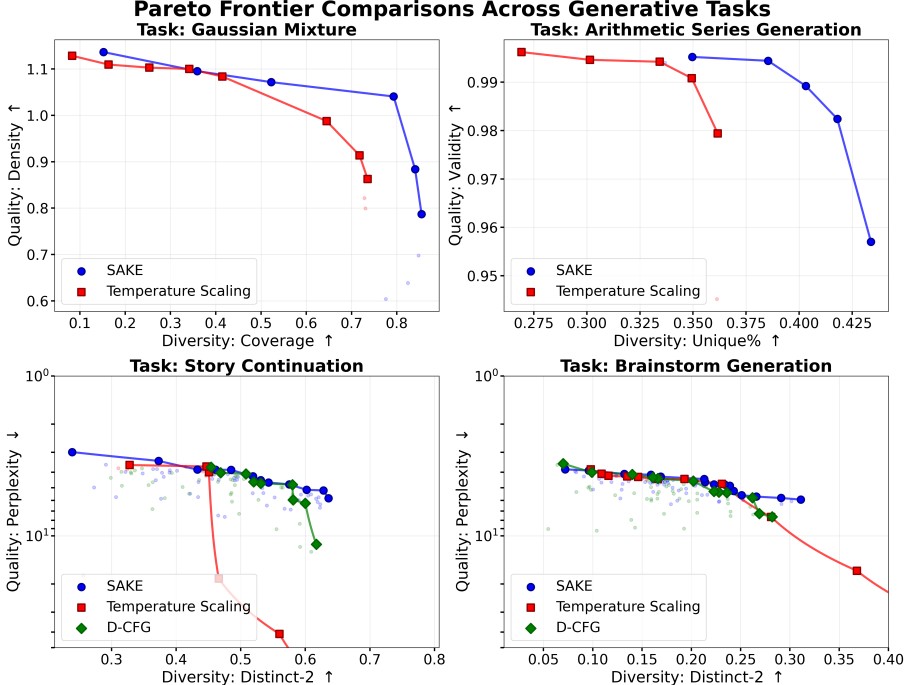

*Figure 2.* Pareto frontier comparisons across three generative tasks, highlighting the diversity–quality trade-off for our diversity guidance and baseline methods. **Top-left:** Gaussian mixture synthesis, plotting coverage vs. density. **Top-right:** arithmetic series generation, plotting uniqueness (unique%) vs. validity. **Bottom:** Two tasks: story continuation and brainstorm generation, plotting distinct-2 vs. perplexity (log scale, lower is better). In each panel, frontier markers summarize the best-achievable quality at a given diversity level. Our SAKE method (blue circles) shifts the frontier outward, improving diversity at comparable quality.

line (23.19 tokens/sec). Conversely, D-CFG experiences a sharp decline to 11.98 tokens/sec, effectively halving the generation speed of the base model. These findings confirm that SAKE provides a lightweight guidance mechanism that scales effectively to longer sequences.

## 6. Limitations and Discussion

Similar to many other training-free guidance methods, our method introduces additional inference-time computation through the evaluation of the semantic kernel and attention kernel. As with guidance-based decoding more broadly, this creates a trade-off between controllability and latency. Although the added cost is substantially smaller than that of guidance baselines, and our embedding-space linearization keeps the complexity linear in sequence length, the method nevertheless incurs non-negligible overhead relative to un-guided decoding. Empirically, the throughput reduction is modest (approximately 7% as shown in Table 3), suggesting that the approach remains practical in many deployment settings. However, this overhead may still limit applicability in real-time scenarios with extremely strict latency constraints, where even small inference slowdowns can be undesirable.

A second limitation concerns the theoretical characterization of guidance-induced distribution shift. While Propo-

sition 4.1 analyzes the effect of guidance on the output distribution through changes in Shannon entropy, this does not yet provide a complete account of how the guidance strength hyperparameter $\gamma$ shapes broader properties of the decoded distribution. In particular, although $\gamma$ can be selected effectively in practice through validation or modest hyperparameter search, a more precise mathematical relationship between guidance strength and target distribution shift remains underdeveloped. A further theoretical treatment of this relationship would improve interpretability and could help reduce empirical tuning.

## 7. Conclusions

We presented a general framework for inference-time diversity guidance in discrete diffusion language models, named *Semantic-Aware Kernel Entropy* (SAKE). By maximizing Rényi entropy within the semantic space, SAKE overcomes the limitations of token-independent assumptions and static temperature scaling. Our method serves as an adaptive modulator, dynamically balancing exploration and coherence to improve algorithmic robustness. Empirical results on code and math reasoning benchmarks confirm that SAKE boosts performance by improving diversity without retraining.

*Table 1.* Qualitative comparison between SAKE and the baseline under identical temperature $T$ and prompt.

| **Prompt:** *"Write 5 prompts for generating a dog image. Be creative."* | |
| --- | --- |
| **Baseline** ($T = 0.2$) | **SAKE** ($T = 0.2, \gamma = 0.7$) |
| 1. A dog running through a field, with a blue sky in the background. 
 2. A dog sitting on a couch, with a blue sky in the background. 
 3. A dog swimming in a lake, with a blue sky in the background. 
 4. A dog playing in a park, with a blue sky in the background. 
 5. A dog sleeping in a bed, with a blue sky in the background. | 1. A golden retriever with a wag on its tail, fetching a ball in a grassy field. 
 2. A white poodle with a mop of fur, walking on a leash in a city park. 
 3. A black labrador with a smile on its face, playing with a toy in a living room. 
 4. A brown husky with a pack on its back, running with a sled in a snowy forest. 
 5. A tan beagle with a bone in its mouth, sitting on a porch in a suburban neighborhood. |

## HumanEval Performance Analysis

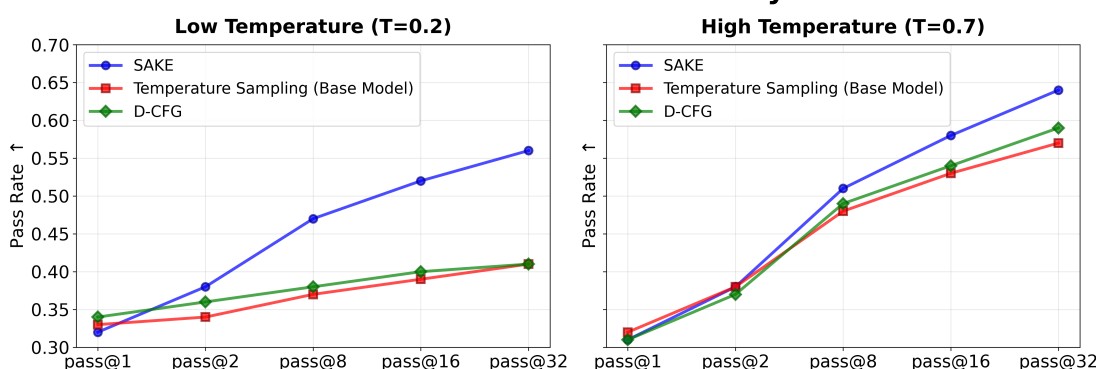

*Figure 3.* Pass@k rates on HumanEval at low ($T = 0.2$) and high ($T = 0.7$) temperatures. At $T = 0.2$, standard Temperature Sampling and D-CFG exhibit signs of mode collapse, while our Entropy Guidance method shows a steady improvement from $k = 1$ to $k = 32$. Notably, our method at $T = 0.2$ achieves a Pass@32 score comparable to the baseline at $T = 0.7$, proving it can extract diverse, high-quality solutions without relying on high temperature.

*Table 2.* Performance comparison on code generation and mathematical reasoning benchmarks.

| **Benchmark** (Metric) | **#Shots** | **Base Model** | **D-CFG** | **SAKE** |
| --- | --- | --- | --- | --- |
| HumanEval (Pass@1) | 0 | 32.9 | **33.7** | 32.0 |
| HumanEval (Pass@32) | 0 | 41.1 | 41.2 | **55.8** |
| MBPP (Pass@1) | 3 | **40.2** | 37.3 | 39.0 |
| MBPP (Pass@32) | 3 | 48.2 | 45.9 | **56.1** |
| GSM8K SC (SM) | 5 | 71.5 | 73.2 | **75.1** |
| GSM8K SC (FE) | 5 | 68.8 | 69.7 | **73.5** |

*Table 3.* Generation speed (Tokens/sec) with standard deviation ($\pm\sigma$).

| | **Prompt Length** | | | | |
| --- | --- | --- | --- | --- | --- |
| **Method** | **128** | **256** | **512** | **1024** | **Average** |
| Base Model | $70.19 \pm 0.42$ | $56.82 \pm 0.34$ | $39.44 \pm 0.06$ | $23.19 \pm 0.02$ | 47.41 |
| D-CFG | $53.41 \pm 1.92$ | $38.42 \pm 0.67$ | $23.53 \pm 0.03$ | $11.98 \pm 0.01$ | 31.83 |
| SAKE | $64.51 \pm 3.45$ | $53.48 \pm 0.21$ | $36.84 \pm 0.14$ | $21.51 \pm 0.01$ | 44.09 |

## Acknowledgment

This work is partially supported by a grant from the Research Grants Council of the Hong Kong Special Administrative Region, China, Project 14210725, and is partially supported by CUHK Direct Research Grant with CUHK Project No. 4055164. The work is also supported by a grant under 1+1+1 CUHK-CUHK(SZ)-GDSTC Joint Collaboration Fund. Also, the authors would like to thank the anonymous reviewers and metareviewer for their constructive suggestions and insightful feedback.

## Impact Statement

This research contributes to the advancement of Machine Learning methodologies. While we acknowledge the broader societal implications inherent to AI technologies, we do not foresee any immediate negative consequences or specific ethical concerns arising directly from this work that necessitate distinct discussion.

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

## A. Proofs

**Proof of Proposition 4.1**

**Proposition A.1** (Entropy Dynamics of Text Diffusion Guidance). *The first-order change in the Shannon entropy of the guided distribution $P_\gamma$ at the limit of zero guidance is governed by the negative covariance between the base log-probabilities and the guidance signal:*

$$\nabla_\gamma H[P_\gamma]\big|_{\gamma=0} = -\mathrm{Cov}_{Y \sim P_{\mathrm{base}}}[\log P_{\mathrm{base}}(Y), \psi_Y]. \tag{19}$$

*Furthermore, since $\log P_{\mathrm{base}}(y) = z_y - \log Z(0)$, this simplifies to:*

$$\nabla_\gamma H[P_\gamma]\big|_{\gamma=0} = -\mathrm{Cov}_{P_{\mathrm{base}}}[\mathbf{z}, \psi]. \tag{20}$$

*Proof.* Let $P_{\mathrm{base}}(y)$ be the probability of a candidate token $y \in \mathcal{V}$ from the vocabulary $\mathcal{V}$ under the unguided base model, defined via logits $z_y$:

$$P_{\mathrm{base}}(y) = \frac{e^{z_y}}{Z(0)},$$

where $Z(0) = \sum_{y' \in \mathcal{V}} e^{z_{y'}}$. Note that $\log P_{\mathrm{base}}(y) = z_y - \log Z(0)$.

The guided distribution $P_\gamma$ is defined by tilting the base distribution with a guidance signal $\psi_y$ scaled by a parameter $\gamma$:

$$P_\gamma(y) = \frac{P_{\mathrm{base}}(y)e^{\gamma\psi_y}}{Z(\gamma)},$$

where the partition function is $Z(\gamma) = \sum_{y' \in \mathcal{V}} P_{\mathrm{base}}(y')e^{\gamma\psi_{y'}} = \mathbb{E}_{Y \sim P_{\mathrm{base}}}[e^{\gamma\psi_Y}]$.

The Shannon entropy of the guided distribution is:

$$H[P_\gamma] = -\sum_y P_\gamma(y) \log P_\gamma(y).$$

We compute the gradient of the entropy with respect to $\gamma$:

$$\nabla_\gamma H[P_\gamma] = \nabla_\gamma \left( -\sum_y P_\gamma(y) \log P_\gamma(y) \right).$$

Using the product rule:

$$\nabla_\gamma H[P_\gamma] = -\sum_y \left( (\nabla_\gamma P_\gamma(y)) \log P_\gamma(y) + P_\gamma(y) \frac{\nabla_\gamma P_\gamma(y)}{P_\gamma(y)} \right).$$

This simplifies to:

$$\nabla_\gamma H[P_\gamma] = -\sum_y (\nabla_\gamma P_\gamma(y)) \log P_\gamma(y) - \sum_y \nabla_\gamma P_\gamma(y).$$

Since $\sum_y P_\gamma(y) = 1$ is constant, its derivative is zero ($\sum_y \nabla_\gamma P_\gamma(y) = 0$). Thus:

$$\nabla_\gamma H[P_\gamma] = -\sum_y (\nabla_\gamma P_\gamma(y)) \log P_\gamma(y). \tag{21}$$

The log-probability is given by:

$$\log P_\gamma(y) = \log P_{\mathrm{base}}(y) + \gamma\psi_y - \log Z(\gamma).$$

Differentiating with respect to $\gamma$:

$$\nabla_\gamma \log P_\gamma(y) = \psi_y - \nabla_\gamma \log Z(\gamma).$$

We compute $\nabla_\gamma \log Z(\gamma)$:

$$\nabla_\gamma \log Z(\gamma) = \frac{1}{Z(\gamma)} \nabla_\gamma \sum_{y' \in \mathcal{V}} P_{\mathrm{base}}(y')e^{\gamma\psi_{y'}} = \sum_{y' \in \mathcal{V}} \frac{P_{\mathrm{base}}(y')e^{\gamma\psi_{y'}}}{Z(\gamma)} \psi_{y'} = \mathbb{E}_{P_\gamma}[\psi].$$

Substituting this back, we get:

$$\nabla_\gamma \log P_\gamma(y) = \psi_y - \mathbb{E}_{P_\gamma}[\psi].$$

Using the identity $\nabla_\gamma P = P(\nabla_\gamma \log P)$:

$$\nabla_\gamma P_\gamma(y) = P_\gamma(y)(\psi_y - \mathbb{E}_{P_\gamma}[\psi]).$$

Substituting the result into Equation (21):

$$\nabla_\gamma H[P_\gamma] = -\sum_y \left[ P_\gamma(y)(\psi_y - \mathbb{E}_{P_\gamma}[\psi]) \right] \log P_\gamma(y)$$

$$= -\sum_y P_\gamma(y)\psi_y \log P_\gamma(y) + \mathbb{E}_{P_\gamma}[\psi] \sum_y P_\gamma(y) \log P_\gamma(y)$$

$$= -\mathbb{E}_{P_\gamma}[\psi_Y \log P_\gamma(Y)] + \mathbb{E}_{P_\gamma}[\psi]\mathbb{E}_{P_\gamma}[\log P_\gamma(Y)].$$

Using the definition of covariance, $\mathrm{Cov}[A, B] = \mathbb{E}[AB] - \mathbb{E}[A]\mathbb{E}[B]$, we obtain:

$$\nabla_\gamma H[P_\gamma] = -\mathrm{Cov}_{Y \sim P_\gamma}[\log P_\gamma(Y), \psi_Y].$$

Taking the limit as $\gamma \to 0$, we have $P_\gamma \to P_{\text{base}}$. Thus:

$$\nabla_\gamma H[P_\gamma]|_{\gamma=0} = -\mathrm{Cov}_{Y \sim P_{\text{base}}}[\log P_{\text{base}}(Y), \psi_Y].$$

This proves Equation (7).

Recall that $\log P_{\text{base}}(y) = z_y - \log Z(0)$. Substituting this into the covariance:

$$\mathrm{Cov}_{P_{\text{base}}}[\log P_{\text{base}}(Y), \psi_Y] = \mathrm{Cov}_{P_{\text{base}}}[z_Y - \log Z(0), \psi_Y].$$

Since covariance is invariant under constant shifts (i.e., $\mathrm{Cov}[X + c, Y] = \mathrm{Cov}[X, Y]$) and $\log Z(0)$ is constant with respect to $Y$:

$$\mathrm{Cov}_{P_{\text{base}}}[z_Y - \log Z(0), \psi_Y] = \mathrm{Cov}_{P_{\text{base}}}[z_Y, \psi_Y].$$

Therefore:

$$\nabla_\gamma H[P_\gamma]|_{\gamma=0} = -\mathrm{Cov}_{P_{\text{base}}}[\mathbf{z}, \psi].$$

This proves Equation (8). $\qquad\qquad\qquad\qquad\qquad\qquad\qquad\qquad\qquad\qquad\qquad\qquad\qquad\qquad\square$

**Proof of Proposition 4.3**

**Proposition A.2** (SAKE Guidance Signal). *Let $\mathbf{H} = [\mathbf{h}_1, \ldots, \mathbf{h}_L] \in \mathbb{R}^{d \times L}$ be the matrix of token embeddings for a sequence of length L. Consider the Gaussian RBF kernel $\kappa(\mathbf{h}_i, \mathbf{h}_j) = \exp(-\|\mathbf{h}_i - \mathbf{h}_j\|^2/2\sigma^2)$ with bandwidth $\sigma > 0$, and let $\mathbf{K} \in \mathbb{R}^{L \times L}$ be the kernel Gram matrix with $\mathbf{K}_{ij} = \kappa(\mathbf{h}_i, \mathbf{h}_j)$. Similarly, $\mathbf{K}_{attn,(i,j)} = \exp(-\|i - j\|^2/2\sigma_{attn}^2)$.*

*The normalized Hadamard product of kernel Gram matrices serves as a valid quantum density matrix. The order-2 Rényi entropy of this representation is:*

$$H_2\left(\frac{1}{L}\mathbf{K} \odot \mathbf{K}_{attn}\right) = -\log\left(\frac{1}{L^2}\|\mathbf{K} \odot \mathbf{K}_{attn}\|_F^2\right). \tag{22}$$

*To increase $H_2$ by adjusting the $i$-th token embedding $\mathbf{h}_i$, we define the continuous diversity function:*

$$\tilde{\mathcal{D}}_\kappa(\mathbf{h}_i; \mathbf{H}_{-i}) \triangleq -\sum_{j \neq i} \kappa_{attn}(i, j)^2 \cdot \kappa(\mathbf{h}_i, \mathbf{h}_j)^2. \tag{23}$$

*where $\mathbf{H}_{-i}$ denotes the embeddings of all tokens except the $i$-th. The gradient of this function with respect to $\mathbf{h}_i$ is:*

$$\nabla_{\mathbf{h}_i} \tilde{\mathcal{D}}_\kappa = \frac{2}{\sigma^2} \sum_{j \neq i} \kappa_{attn}(i, j)^2 \cdot \kappa(\mathbf{h}_i, \mathbf{h}_j)^2 (\mathbf{h}_i - \mathbf{h}_j). \tag{24}$$

We verify the proposition in three parts: (1) the validity of the density matrix, (2) the derivation of the Rényi entropy, and (3) the gradient of the diversity function.

Let $\mathbf{M} = \mathbf{K} \odot \mathbf{K}_{\text{attn}}$. We claim that $\rho = \frac{1}{L}\mathbf{M}$ is a valid density matrix. A matrix $\rho$ is a density matrix if it is Hermitian, Positive Semi-Definite (PSD), and has unit trace.

1. **Hermitian Property:** Both $\mathbf{K}$ and $\mathbf{K}_{\text{attn}}$ are real symmetric matrices (since the Gaussian kernel $\kappa(x, y) = \kappa(y, x)$ is symmetric). The Hadamard product of two symmetric matrices is symmetric. Thus, $\rho$ is real and symmetric, which implies it is Hermitian ($\rho = \rho^\dagger$).

2. **Positive Semi-Definite (PSD):** $\mathbf{K}$ is a Gram matrix generated by a valid kernel (Gaussian RBF), so $\mathbf{K} \succeq 0$. Similarly, $\mathbf{K}_{\text{attn}}$ is a Gram matrix generated by a Gaussian kernel over indices, so $\mathbf{K}_{\text{attn}} \succeq 0$. By the **Schur Product Theorem**, the Hadamard product of two PSD matrices is also PSD. Therefore, $\mathbf{M} \succeq 0$ and $\rho \succeq 0$.

3. **Unit Trace:** The diagonal elements of a Gaussian kernel Gram matrix are all 1, because $\kappa(\mathbf{x}, \mathbf{x}) = \exp(0) = 1$. Thus, $\mathbf{M}_{ii} = \mathbf{K}_{ii} \cdot (\mathbf{K}_{\text{attn}})_{ii} = 1 \cdot 1 = 1$. The trace is:

$$\text{Tr}(\rho) = \text{Tr}\left(\frac{1}{L}\mathbf{M}\right) = \frac{1}{L}\sum_{i=1}^{L}\mathbf{M}_{ii} = \frac{1}{L} \cdot L = 1.$$

Therefore, $\rho = \frac{1}{L}\mathbf{K} \odot \mathbf{K}_{\text{attn}}$ is a valid density matrix.

The definition of Order-2 Rényi entropy for a density matrix $\rho$ is:

$$H_2(\rho) = -\log(\text{Tr}(\rho^2)).$$

Substituting $\rho = \frac{1}{L}\mathbf{M}$:

$$\text{Tr}(\rho^2) = \text{Tr}\left(\left(\frac{1}{L}\mathbf{M}\right)^2\right) = \frac{1}{L^2}\text{Tr}(\mathbf{M}^2).$$

Since $\mathbf{M}$ is real and symmetric, $\mathbf{M}^\top = \mathbf{M}$. The trace of the square of a symmetric matrix is the sum of the squares of its elements, which is the squared Frobenius norm:

$$\text{Tr}(\mathbf{M}^2) = \text{Tr}(\mathbf{M}^\top\mathbf{M}) = \|\mathbf{M}\|_F^2.$$

Substituting this back into the entropy equation:

$$H_2(\rho) = -\log\left(\frac{1}{L^2}\|\mathbf{M}\|_F^2\right) = -\log\left(\frac{1}{L^2}\|\mathbf{K} \odot \mathbf{K}_{\text{attn}}\|_F^2\right).$$

This matches Equation 22 in the proposition.

Maximizing the entropy $H_2(\rho)$ is equivalent to minimizing the argument of the logarithm, specifically the squared Frobenius norm $\|\mathbf{K} \odot \mathbf{K}_{\text{attn}}\|_F^2$. This norm represents the collision probability or "purity" of the state.

To derive a tractable diversity signal for guiding the $i$-th token, we decompose the Frobenius norm of the Hadamard product matrix $\mathbf{M} = \mathbf{K} \odot \mathbf{K}_{\text{attn}}$. The elements are $\mathbf{M}_{jk} = \kappa(\mathbf{h}_j, \mathbf{h}_k) \cdot \kappa_{\text{attn}}(j, k)$.

$$\|\mathbf{M}\|_F^2 = \sum_{j=1}^{L}\sum_{k=1}^{L}\left(\kappa_{\text{attn}}(j, k) \cdot \kappa(\mathbf{h}_j, \mathbf{h}_k)\right)^2 \tag{25}$$

$$= \underbrace{\sum_{j\neq i}\sum_{k\neq i}\mathbf{M}_{jk}^2}_{\text{independent of }\mathbf{h}_i} + \underbrace{\mathbf{M}_{ii}^2}_{\text{diagonal term}} + 2\underbrace{\sum_{j\neq i}\mathbf{M}_{ij}^2}_{\text{cross terms}}. \tag{26}$$

We observe that the diagonal term $\mathbf{M}_{ii} = \kappa(\mathbf{h}_i, \mathbf{h}_i)\kappa_{\mathrm{attn}}(i, i) = 1 \cdot 1 = 1$, so $\mathbf{M}_{ii}^2 = 1$, which is constant. The cross terms involve the interaction between token $i$ and all other tokens $j$:

$$2\sum_{j \neq i} \mathbf{M}_{ij}^2 = 2\sum_{j \neq i} \left(\kappa_{\mathrm{attn}}(i,j) \cdot \kappa(\mathbf{h}_i, \mathbf{h}_j)\right)^2 = 2\sum_{j \neq i} \kappa_{\mathrm{attn}}(i,j)^2 \cdot \kappa(\mathbf{h}_i, \mathbf{h}_j)^2.$$

Minimizing $\|\mathbf{M}\|_F^2$ with respect to $\mathbf{h}_i$ is therefore equivalent to minimizing the sum of these squared cross-terms. This motivates the definition of the continuous diversity function $\tilde{\mathcal{D}}_\kappa$, which we seek to *maximize* (due to the negative sign):

$$\tilde{\mathcal{D}}_\kappa(\mathbf{h}_i; \mathbf{H}_{-i}) \triangleq -\sum_{j \neq i} \kappa_{\mathrm{attn}}(i,j)^2 \cdot \kappa(\mathbf{h}_i, \mathbf{h}_j)^2.$$

We compute the gradient of the function defined in Equation (2):

$$\tilde{\mathcal{D}}_\kappa(\mathbf{h}_i) = -\sum_{j \neq i} C_{ij} \cdot \kappa(\mathbf{h}_i, \mathbf{h}_j)^2,$$

where $C_{ij} = \kappa_{\mathrm{attn}}(i,j)^2$.

First, recall the definition of the kernel $\kappa(\mathbf{h}_i, \mathbf{h}_j) = \exp\left(-\frac{\|\mathbf{h}_i - \mathbf{h}_j\|^2}{2\sigma^2}\right)$. The squared kernel is:

$$\kappa(\mathbf{h}_i, \mathbf{h}_j)^2 = \exp\left(-\frac{\|\mathbf{h}_i - \mathbf{h}_j\|^2}{\sigma^2}\right).$$

Applying the chain rule to find the gradient with respect to $\mathbf{h}_i$:

$$\nabla_{\mathbf{h}_i}\left(\kappa(\mathbf{h}_i, \mathbf{h}_j)^2\right) = \nabla_{\mathbf{h}_i} \exp\left(-\frac{\|\mathbf{h}_i - \mathbf{h}_j\|^2}{\sigma^2}\right)$$

$$= \exp\left(-\frac{\|\mathbf{h}_i - \mathbf{h}_j\|^2}{\sigma^2}\right) \cdot \nabla_{\mathbf{h}_i}\left(-\frac{\|\mathbf{h}_i - \mathbf{h}_j\|^2}{\sigma^2}\right)$$

$$= \kappa(\mathbf{h}_i, \mathbf{h}_j)^2 \cdot \left(-\frac{1}{\sigma^2} \cdot 2(\mathbf{h}_i - \mathbf{h}_j)\right)$$

$$= -\frac{2}{\sigma^2}\kappa(\mathbf{h}_i, \mathbf{h}_j)^2(\mathbf{h}_i - \mathbf{h}_j).$$

Substituting this into the gradient of the diversity function:

$$\nabla_{\mathbf{h}_i}\tilde{\mathcal{D}}_\kappa = -\sum_{j \neq i} C_{ij} \cdot \nabla_{\mathbf{h}_i}\left(\kappa(\mathbf{h}_i, \mathbf{h}_j)^2\right)$$

$$= -\sum_{j \neq i} C_{ij} \cdot \left(-\frac{2}{\sigma^2}\kappa(\mathbf{h}_i, \mathbf{h}_j)^2(\mathbf{h}_i - \mathbf{h}_j)\right)$$

$$= \frac{2}{\sigma^2}\sum_{j \neq i} \kappa_{\mathrm{attn}}(i,j)^2 \cdot \kappa(\mathbf{h}_i, \mathbf{h}_j)^2(\mathbf{h}_i - \mathbf{h}_j).$$

$\square$

## B. Detailed Analysis of Gaussian Synthesis

To demonstrate the distinct behaviors of temperature scaling versus our proposed diversity guidance, we first examine a controlled environment with a known ground truth. Specifically, we use a simple discrete diffusion model to approximate a 2D Gaussian mixture distribution, a setting where high-probability modes are explicitly defined. This allows for a precise evaluation of generation quality and diversity based on mode coverage.

Figure 4 illustrates the synthesis of a 8-component Gaussian mixture target. The top row presents an unguided baseline where the key hyperparameter is the sampling temperature $T \in \{1, 5, 10, 20\}$, which modulates the stochasticity of the

reverse dynamics. Low $T$ tends to yield overly concentrated samples with limited mode coverage, while higher $T$ increases exploration and covers more mixture components. However, this improved coverage comes at a clear cost in sample quality. When $T$ is large, the particle clouds become substantially more diffuse and less well-aligned with individual Gaussian components, indicating degraded fidelity despite broader support. The bottom row applies our proposed diversity guidance term with strength $\gamma \in \{0.1, 0.2, 0.5, 1.0\}$, showing progressively improved mode coverage as $\gamma$ increases. Particles distribute across the mixture components more evenly and visually better match the target support. These results suggest that while temperature scaling forces a trade-off between concentration and exploration, our proposed method effectively regularizes against mode collapse, enabling broad coverage without relying on high-temperature sampling.

## C. Detailed Analysis of Diversity-Quality Trade-off

We extend our evaluation to broader settings by analyzing the Diversity-Quality Pareto frontier. The Pareto frontier reveals the optimal trade-off boundary between two competing objectives. In this context, an outward shift of the frontier indicates that a method achieves higher diversity for a given quality budget (or vice versa).

Figure 2 demonstrates that *diversity-guided* sampling consistently yields superior Pareto frontiers compared to standard *temperature sampling* across three distinct generative tasks. In all settings, we observe that while temperature sampling forces a strict trade-off that sacrificing quality for diversity, diversity guidance mitigates this degradation, maintaining higher sample quality at comparable levels of diversity.

**Gaussian Mixture Generation (Fig. 2, top-left)**: We evaluate the trade-off between Coverage (diversity metric) and Density (quality metric) proposed by Naeem et al. (2020), computed using $k=5$ nearest neighbors and a bandwidth of $\sigma=3$ over 2,000 samples. The generation process involves 500 steps with 2,000 particles initialized from a single point. While varying temperature $T \in [1, 30]$ expands coverage, it causes a precipitous decline in density. Conversely, sweeping the guidance strength $\gamma \in [0.01, 2.0]$ shifts the frontier outward, achieving higher coverage for equivalent density levels.

**Arithmetic Series Generation (Fig. 2, top-right)**: We select this task as a controlled setting where quality and diversity metrics are mathematically precise and unambiguous. The objective is to generate number sequences that follow a strict arithmetic progression. We measure *Validity* (the fraction of generated sequences satisfying the arithmetic constraint) against *Uniqueness* (the fraction of unique sequences among valid samples). Results are averaged over 5 runs of 1,000 samples each. We observe that high-temperature sampling improves uniqueness only at the cost of a sharp drop in validity. In contrast, diversity guidance significantly extends the Pareto frontier, allowing for high uniqueness with minimal loss in validity.

**Story Continuation (Fig. 2, bottom-left)**: We analyze the trade-off between Perplexity (quality metric; lower is better) and Distinct-2 (diversity metric) proposed by Li et al. (2016). We employ a discrete diffusion language model LLaDA (Nie et al., 2025) for generation and Qwen3-4B (Yang et al., 2025) as the external evaluator. The results indicate that aggressive temperature sampling incurs a steep perplexity penalty for marginal gains in diversity. Diversity guidance attains a more favorable balance, achieving higher Distinct-2 scores at lower perplexity levels. While Classifier-Free Guidance (CFG) offers a competitive baseline in specific regimes, diversity guidance maintains robust performance in high-diversity ranges.

**Brainstorm Generation (Fig.2, bottom-right)**: We design this experiment to require the model to generate multiple proposals for a single prompt within one generation. For example, "*Generate 5 prompts for an image of dog. Be creative.*" We use the same generative model and evaluator as in the story continuation task. We observe that this task presents a challenging trade-off for baseline methods. As shown by the red curve, temperature sampling suffers from a rapid degradation in quality (increasing perplexity) as it attempts to increase diversity beyond a Distinct-2 score of 0.25. Similarly, D-CFG (green diamonds) remains clustered in a conservative region, failing to explore the high-diversity distinctness required for brainstorming. In contrast, diversity guidance establishes a superior Pareto frontier in the high-diversity region, maintaining low perplexity (high quality) even as the Distinct-2 metric increases. The method effectively decouples diversity from significant quality loss, allowing the model to generate varied, creative ideas without sacrificing semantic coherence. A case study comparing the baseline and our SAKE method under the same temperature setting further illustrates this behavior. As shown in Table1, the baseline collapses into repetitive syntactic templates, recycling identical phrases across items. Conversely, SAKE successfully diversifies the semantic content, varying subjects (breeds), actions, and settings while maintaining coherence.

*Table 4.* Sensitivity to semantic bandwidth $\sigma$ for MDLM. Lower Gen PPL is better.

| MDLM ($\sigma$) | 5 | 10 | 20 | 30 | 40 | 80 | 120 |
|---|---|---|---|---|---|---|---|
| **Gen PPL** $\downarrow$ | $10.69 \pm 1.69$ | $10.43 \pm 1.17$ | $10.13 \pm 0.81$ | $10.78 \pm 1.03$ | $11.94 \pm 1.02$ | $16.53 \pm 1.39$ | $18.33 \pm 2.08$ |

*Table 5.* Sensitivity to semantic bandwidth $\sigma$ for BD3LM. Lower Gen PPL is better.

| BD3LM ($\sigma$) | 40 | 60 | 80 | 100 | 120 | 200 | 300 |
|---|---|---|---|---|---|---|---|
| **Gen PPL** $\downarrow$ | $43.69 \pm 2.69$ | $42.43 \pm 2.17$ | $42.14 \pm 1.39$ | $43.13 \pm 2.31$ | $44.39 \pm 2.42$ | $51.65 \pm 3.65$ | $57.11 \pm 2.18$ |

# D. Additional Numerical Results

**LLaDA on HumanEval with Nucleus Sampling (top-p).** We further examine the effect of nucleus sampling on code-generation performance by sweeping *top-p* values for the base LLaDA model on HumanEval. Table 8 reports the corresponding `pass@32` results. Performance improves substantially as *top-p* increases from 0.5 to 0.7, rising from 35.4 to 40.9. Beyond this range, performance remains relatively stable, with values between 39.6 and 41.2 across *top-p* settings from 0.8 to 1.0. The best result is obtained at *top-p* = 0.95, which achieves a `pass@32` of 41.2. This observation suggests that moderate-to-high sampling diversity is beneficial for HumanEval, while overly restrictive sampling degrades performance. Overall, these results support the use of the standard HumanEval setting of *top-p* = 0.95 in our experiments.

**Hyperparameter sensitivity.** We further examine the sensitivity of our method to the two main bandwidth hyperparameters: the semantic bandwidth $\sigma$ and the attention bandwidth $\sigma_{\text{attn}}$. Overall, the results indicate that performance is stable across broad hyperparameter ranges, suggesting that the method does not require fine-grained tuning.

**Semantic bandwidth $\sigma$.** The semantic bandwidth $\sigma$ determines the similarity scale at which differences between token representations are treated as meaningful. Smaller values of $\sigma$ make the method more sensitive to local semantic variation, while larger values impose a broader notion of diversity. Intuitively, smaller $\sigma$ values may be better suited for discouraging near-duplicate generations or repetitive loops, whereas larger values can encourage diversity at the level of broader semantic themes.

Tables 4 and 5 report the sensitivity of generation perplexity (Gen PPL) to $\sigma$ for MDLM (Sahoo et al., 2024) and BD3LM (Arriola et al., 2025), respectively. For MDLM, the best performance is achieved around $\sigma = 20$, but results remain close to optimal throughout the range $[5, 30]$. For BD3LM, the lowest Gen PPL is obtained near $\sigma = 80$, with similarly stable performance over the broader interval $[40, 120]$. In both cases, performance degrades only when $\sigma$ becomes substantially larger than these ranges. These findings suggest that the semantic bandwidth is robust to moderate misspecification, and that selecting $\sigma$ within a broad reasonable interval is sufficient in practice.

**Attention bandwidth $\sigma_{\text{attn}}$.** The attention bandwidth $\sigma_{\text{attn}}$ controls the effective positional receptive field by modulating how quickly attention decays with token distance. Larger values permit stronger influence from more distant tokens, whereas smaller values concentrate attention more locally.

A useful interpretation of $\sigma_{\text{attn}}$ can be obtained by defining a small threshold $\epsilon$ at which attention is considered to have effectively vanished. For token distance $d$, this yields

$$\exp\left(-\frac{d^2}{2\sigma_{\text{attn}}^2}\right) = \epsilon \quad \implies \quad d = \sigma_{\text{attn}}\sqrt{2\ln\left(\frac{1}{\epsilon}\right)}. \tag{27}$$

For example, when $\epsilon = 0.01$, the corresponding vanishing distance is approximately $d_v \approx 3.03\,\sigma_{\text{attn}}$. This provides a simple initialization heuristic: one may choose $\sigma_{\text{attn}}$ by specifying a desired effective context range, thereby substantially reducing the need for exhaustive tuning.

Tables 6 and 7 summarize the empirical sensitivity of Gen PPL to $\sigma_{\text{attn}}$. For BD3LM, performance improves substantially as $\sigma_{\text{attn}}$ increases from very small values, and remains near-optimal across a broad range from 10 to 80. For MDLM, the best values are obtained around $\sigma_{\text{attn}} = 20$ to 40, but performance remains stable over a much wider interval extending up to 300. Taken together, these results indicate that $\sigma_{\text{attn}}$ is also not highly sensitive, and that the vanishing-distance heuristic provides a practical and interpretable guideline for setting this parameter.

*Table 6.* Sensitivity to attention bandwidth $\sigma_{\text{attn}}$ for BD3LM. Lower Gen PPL is better.

| BD3LM ($\sigma_{\text{attn}}$) | 1 | 2 | 5 | 10 | 20 | 40 | 80 |
|---|---|---|---|---|---|---|---|
| **Gen PPL** ↓ | $52.44 \pm 1.83$ | $47.17 \pm 1.84$ | $43.73 \pm 1.43$ | $42.35 \pm 1.90$ | $42.14 \pm 1.39$ | $43.57 \pm 1.71$ | $42.52 \pm 1.98$ |

*Table 7.* Sensitivity to attention bandwidth $\sigma_{\text{attn}}$ for MDLM. Lower Gen PPL is better.

| MDLM ($\sigma_{\text{attn}}$) | 10 | 20 | 40 | 80 | 120 | 200 | 300 |
|---|---|---|---|---|---|---|---|
| **Gen PPL** ↓ | $11.10 \pm 1.48$ | $10.13 \pm 0.81$ | $10.02 \pm 1.89$ | $10.77 \pm 0.98$ | $11.05 \pm 1.15$ | $11.24 \pm 0.31$ | $11.37 \pm 1.13$ |

## 9 Gaussian Mixture Synthesis for Discrete Diffusion Reverse Process

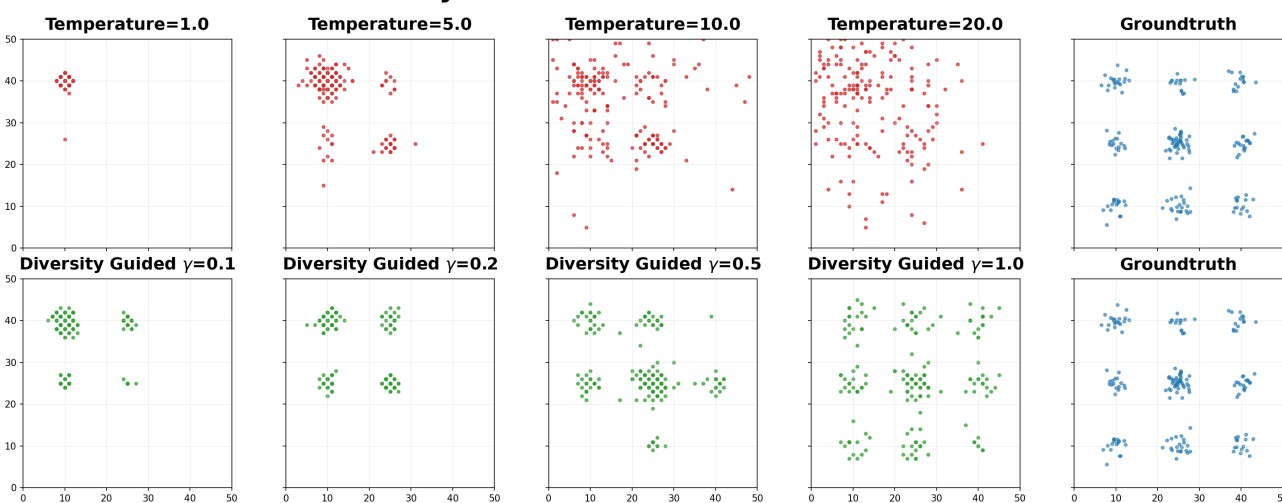

*Figure 4.* Comparison of sampling behaviors for a 2D 9-Gaussian-mixture target. **Top row**: baseline unguided reverse process at different temperatures $T \in \{1, 5, 10, 20\}$, illustrating a coverage-fidelity trade-off (higher $T$ covers more modes but yields more diffuse, lower-quality samples). **Bottom row**: Diversity guidance with strengths $\gamma \in \{0.1, 0.2, 0.5, 1.0\}$, which improves mode coverage while maintaining better precision. Rightmost column shows ground-truth samples.

*Table 8.* Effect of nucleus sampling (*top-p*) on HumanEval for the base LLaDA model.

| LLaDA (*top-p*) | 0.5 | 0.6 | 0.7 | 0.8 | 0.9 | 0.95 | 1.0 |
|---|---|---|---|---|---|---|---|
| **HumanEval pass@32** | 35.4 | 36.6 | 40.9 | 40.9 | 40.2 | **41.2** | 39.6 |

*Table 9.* Comparison of pass@k results across HumanEval and MBPP benchmarks, with varying temperatures.

| Benchmark | Model | Temp. | pass@1 | pass@2 | pass@8 | pass@16 | pass@32 |
|---|---|---|---|---|---|---|---|
| HumanEval | baseline | 0.2 | 0.33 | 0.34 | 0.37 | 0.39 | 0.41 |
| | D-CFG | 0.2 | 0.34 | 0.36 | 0.38 | 0.40 | 0.41 |
| | SAKE | 0.2 | 0.32 | 0.38 | 0.47 | 0.52 | 0.56 |
| | baseline | 0.5 | 0.33 | 0.37 | 0.46 | 0.50 | 0.53 |
| | D-CFG | 0.5 | 0.32 | 0.38 | 0.47 | 0.51 | 0.55 |
| | SAKE | 0.5 | 0.32 | 0.38 | 0.49 | 0.54 | 0.59 |
| | baseline | 0.7 | 0.32 | 0.38 | 0.48 | 0.53 | 0.57 |
| | D-CFG | 0.7 | 0.31 | 0.37 | 0.49 | 0.54 | 0.59 |
| | SAKE | 0.7 | 0.31 | 0.38 | 0.51 | 0.58 | 0.64 |
| MBPP | baseline | 0.2 | 0.40 | 0.42 | 0.46 | 0.47 | 0.48 |
| | D-CFG | 0.2 | 0.37 | 0.41 | 0.43 | 0.44 | 0.46 |
| | SAKE | 0.2 | 0.39 | 0.44 | 0.51 | 0.54 | 0.56 |

