# OpenReview forum: "Exploring More to Solve More: Boosting Diversity in Text Diffusion Models via Entropy-Based Guidance"
_ICML.cc/2026/Conference — ICML 2026 regular_

### Official Review · Reviewer_SDvU · 2026-03-12

**Soundness:** 3
**Presentation:** 2
**Significance:** 2
**Originality:** 3
**Overall Recommendation:** 4
**Confidence:** 4

**Summary:**

This paper aims to address the challenge faced by Discrete Text Diffusion Models in balancing "fidelity" and "diversity" during the generation process. Existing sampling strategies (such as temperature scaling or D-CFG) mostly adjust token probabilities without semantic awareness, leading to compromised generation quality. To tackle this, the authors propose a training-free Semantic-Aware Kernel Entropy guidance mechanism (SAKE). This method linearizes objectives in the embedding space to dynamically adjust the generation distribution, encouraging exploration when needed while maintaining high fidelity when diversity is achieved.

**Compliance With Llm Reviewing Policy:**

Affirmed.

**Final Justification:**

I thank the authors for their clarifications. All of my concerns have been addressed, I will raise my score and recommend to acepti

**Key Questions For Authors:**

Please see Weaknesses.

**Limitations:**

yes

**Strengths And Weaknesses:**

**Strengths**
1. The paper proposes a generalized discrete text diffusion diversity-guided framework. Through Proposition 4.1, the authors mathematically prove that the covariance between the guidance signal and the baseline distribution (base log-probabilities) determines the increase or decrease of the Shannon entropy in the generative system.
2. raditional guidance in discrete state spaces often requires calculating an exponential-level partition function. By linearizing the diversity function in a continuous embedding space, the authors reduce the computational complexity to only require one backpropagation, greatly improving computational efficiency.

**Weaknesses**
1. As the author admitted in the conclusion, calculating the semantic kernel and attention kernel will inevitably introduce a certain amount of inference time overhead. This may limit the application of this method in real-time application scenarios with extremely high latency requirements.
2. The precise mathematical relationship between the guidance strength hyperparameters in the current method (e.g., obtained through grid search in experiments) and the target distribution shift still lacks in-depth theoretical justification and characterization.
3. This paper lacks qualitative comparisons, making it difficult to intuitively reflect changes in image diversity.

---

> ### Author Rebuttal · Authors · 2026-03-31
>
> We sincerely thank the reviewer for their constructive and insightful feedback. Below is our response to the comments and questions in the review.
>
> **Q1: Computation Efficiency**
>
> We thank the reviewer for highlighting the consideration of inference latency. It is true that applying guidance introduces an additional step to steer the diffusion model's outputs, resulting in some inference overhead. However, this trade-off allows practitioners to finely control model outputs without the computational costs and time required to retrain or fine-tune the model.
>
> Furthermore, we have minimized this overhead, making it substantially lower than that of baseline guidance methods. As detailed in our methodology section, our embedding-space linearization reduces the computational complexity to be linear with respect to sequence length, ensuring strong scalability. Empirically, as shown in Section 5.4, SAKE introduces only a marginal throughput drop of approximately 7%. The majority of the latency remains bounded by the large language model's standard forward pass, making our method highly practical for most deployment scenarios.
>
> **Q2: Relationship between Guidance and Distribution Shift**
>
> In proposition 4.1, we provide an analysis on how temperature scaling and guidance methods affect the probability distribution, specifically focusing on the Shannon entropy change to quantify this shift. The guidance strength hyperparameter $\gamma$ controls the shift amount. We believe theoretical analysis of broader range of distribution shift is a highly valuable contribution, and could be an important direction for future theoretical work.
>
> **Q3: Qualitative Samples and Image Diversity**
>
> Regarding the mention of "image diversity," we would like to kindly clarify that our method and experiments are specifically designed for text generation and diffusion language models, rather than image generation.
>
> For qualitative comparison, we have provided a table in the current draft showing qualitative comparisons between the baseline and our method. To further address your feedback, we will gladly expand this section and include a broader range of qualitative text samples in the appendix of the revised manuscript to better illustrate the diversity gains.

---

> > ### Author Rebuttal · Reviewer_SDvU · 2026-04-04
> >
> > please see my new comment

---

> > > ### Author Response · Authors · 2026-04-05
> > >
> > > Dear Reviewer SDvU,
> > >
> > > We would like to kindly note that the comment you referred to (“please see my new comment”) is currently not visible to us on the authors’ side. It may have been posted as a private comment visible only to the Reviewers/AC. We would greatly appreciate it if you could make the comment visible to the authors, so that we can properly address your points.
> > >
> > > Best regards,\
> > > Authors

---

### Official Review · Reviewer_Gnea · 2026-03-13

**Soundness:** 3
**Presentation:** 3
**Significance:** 3
**Originality:** 3
**Overall Recommendation:** 5
**Confidence:** 3

**Summary:**

This paper explores sampling in Diffusion Language Models (DLMs) with the goal of pushing the fidelity-diversity trade-off boundary through an entropy-based guidance approach. The authors note that guidance in diffusion models requires different strategies in continuous domains (like images) compared to discrete domains (like language) due to differentiability constraints and the fact that the logit space in text diffusion lacks the semantic structure present in continuous feature spaces. Challenging the traditional view that diversity compromises quality, the authors argue that in complex reasoning, generating diverse chains of thought actually improves robustness and prevents the model from collapsing into a single incorrect argument. To implement this, they propose SAKE, an objective that models diversity by acting as an adaptive modulator that flattens or sharpens the generation distribution based on the semantic redundancy of the context. The method relies on a guidance vector $\psi$, which effectively acts as a logit over the vocabulary to shift the base model's probability. Rather than a brute-force evaluation of the diversity "gap" between tokens, the core innovation involves a first-order Taylor expansion to approximate $\psi$ via the gradient of a diversity function. This diversity function is built from kernel functions of token embedding vectors to efficiently capture both semantic similarity and relative token positions.

**Compliance With Llm Reviewing Policy:**

Affirmed.

**Key Questions For Authors:**

1. Could you clarify Equation (13)? It would be helpful if the authors could provide additional explanation on how this expression is obtained.

2. What is the effect of $\sigma$ and $\sigma_{attn}$ in the kernels in the performance? How are they tuned?

**Limitations:**

yes

**Strengths And Weaknesses:**

### Strengths

1. Intuitive and well-motivated method : The proposed approach is conceptually intuitive, and its theoretical formulation appears sound. The main propositions are clearly stated and supported with rigorous analysis.

2. Clear presentation of the technical framework: The paper does a good job of presenting the background, preliminaries, and method in a structured and accessible manner, making the overall approach easy to follow.

3. Original treatment of diversity in diffusion language models : The paper offers a novel perspective on the fidelity–diversity trade-off by introducing an entropy-based diversity objective tailored to diffusion language models. This gives the work a clear element of originality.

### Weaknesses
1. The adaptive guidance mechanism could be explained more clearly. While the high-level intuition is appealing, the paper would benefit from a more explicit explanation of how the adaptive guidance behaves in the two opposite regimes—when it flattens the distribution versus when it sharpens it. Clarifying the conditions under which each behavior emerges would make the method easier to interpret.

2. The justification for the linearization in Equation (13) is insufficient. It is not fully clear why the first-order approximation used in Equation (13) is valid in this setting. The paper should better explain the assumptions under which this linearization is expected to hold, and ideally discuss its accuracy or limitations in practice.

---

> ### Author Rebuttal · Authors · 2026-03-31
>
> We sincerely thank the reviewer for their constructive and insightful feedback. Below is our response to the comments and questions in the review.
>
> **Q1: Adaptive Guidance Mechanism**
>
> Per Proposition 4.1, SAKE's adaptability depends on the covariance between base logits $z$ and the diversity signal $\psi$
>
> - **Flattening (Exploration)**: If the base model experiences mode collapse during reverse diffusion (probability mass concentrated on a few redundant modes), $\psi$ assigns high scores to underexplored, semantically distinct tokens while penalizing the collapsed mode. This creates a negative covariance between $z$ and $\psi$, and increases the entropy of the guided distribution to encourage exploration.
>
> - **Sharpening (Exploitation)**: Conversely, if the base distribution is already diverse, guidance signals minimize (Eq. 18: higher diversity yields lower kernel values). The signal $\psi$ becomes uniform or aligns with the model's confidence. Covariance approaches zero or turns positive, letting base logits dominate. This sharpens the distribution around valid tokens, preserving fidelity.
>
> To illustrate this, we tracked the guidance signal and entropy during sampling ($T \in [0,1]$, ending at $T=1$). We observe three phases: (1) **Exploration ($T \in [0, 0.3]$)**: positive entropy change ($\Delta H>0$), and strong guidance. (2) **Exploitation ($T \in [0.4, 0.6]$)**: negative entropy change, moderate signal. (3) **Vanishing**: sequences are diversified, signal vanishes.
>
> | T | 0.1 | 0.2 | 0.3 | 0.4 | 0.5 | 0.6 | 0.7 | 0.8 |
> |-|-|-|-|-|-|-|-|-|
> | $\Vert \psi \Vert$ | 6.75 | 2.86 | 3.50 | 1.72 | 0.80 | 0.49 | 0.11 | 0.04 |
> | $\Delta H$ | 0.15 | 0.07 | 0.11 | -0.03 | -0.07 | -0.09 | 0.01 | <0.01 |
>
> We will revised Section 4.2 to better explain this dynamic.
>
> ---
> **Q2: Clarification of Eq. 13**
>
> Eq. 13 relies on a first-order Taylor expansion, assuming a smooth continuous diversity function $\tilde{D}$ and bounded embedding distances $\Vert \mathbf{h_y} - \mathbf{h_{x_i}}\Vert$. These assumptions hold in modern LLMs (like LLaDA) because LayerNorm/RMSNorm constrains embeddings to a hypersphere, and Gaussian kernel in our $\tilde{D}$ is smooth.
>
> A theoretical limitation of this linear approximation is that, unlike the true bounded RBF kernel, it does not plateau. Under large guidance scales (𝛾), an outlier token lying far along the gradient could receive an overestimated diversity score.
>
> Fortunately, this edge case is controllable. By restricting the guidance signal to a 'trust region', simply applying a mild Top-p or Top-k filter to the base logits before guidance, we ensure diversification only occurs among tokens that are semantically plausible. We will clarify this in the revision.
>
> ---
> **Q3: Hyperparameters**
>
> **Semantic Bandwidth 𝜎**:
>
> 𝜎 sets the similarity detection threshold. Small 𝜎 values are sensitive to diversity (e.g., penalizing at synonyms level), while large 𝜎 enforce broader diversity. In practice, small 𝜎 could prevent repetition or looping, while large 𝜎 may encourage topic-level diversity for creative writing.
>
> Gen PPL evaluations demonstrate that optimal 𝜎 values are stable across broad ranges:
>
> | MDLM (𝜎) | 5 | 10 | 20 | 30 | 40 | 80 | 120 |
> |--|-|-|-|-|-|-|-|
> | Gen PPL ↓ | 10.69 ± 1.69 | 10.43 ± 1.17 | 10.13 ± 0.81 | 10.78 ± 1.03 | 11.94 ± 1.02 | 16.53 ± 1.39 | 18.33 ± 2.08 |
>
> | BD3LM (𝜎) | 40 | 60 | 80 | 100 | 120 | 200 | 300
> |--|-|-|-|-|-|-|-|
> | Gen PPL ↓ | 43.69 ± 2.69 | 42.43 ± 2.17 | 42.14 ± 1.39 | 43.13 ± 2.31 | 44.39 ± 2.42 | 51.65 ± 3.65 | 57.11 ± 2.18 |
>
> MDLM is near-optimal for 𝜎 $\in$ [5, 30] and BD3LM for 𝜎 $\in$ [40, 120]. Thus, fine-grained tuning is unnecessary. Selecting 𝜎 within a broad reasonable range suffices for gains.
>
> **Attention Bandwidth 𝜎_attn**:
>
> 𝜎_attn controls the positional receptive field. Larger 𝜎_attn allow influence from more distant tokens.
>
> We offer an interpretable initialization for 𝜎_attn. For a token distance $d$, define a threshold $\epsilon$ where attention vanishes:
>
> $$
> \exp\left(-\frac{d^2}{2\sigma_{\text{attn}}^2}\right) = \epsilon \implies d = \sigma_{\text{attn}} \sqrt{2 \ln\left(\frac{1}{\epsilon}\right)}
> $$
>
> If attention vanishes at 1% $(\epsilon=0.01)$, the vanishing distance is $d_v \approx 3.03 𝜎_{attn}$. Practitioners can set 𝜎_attn based on a desired vanishing distance (e.g. context length), minimizing tuning. Empirical sensitivity analysis shows stable optimal performance across a wide range of 𝜎_attn:
>
> | BD3LM (𝜎_attn) | 1 | 2 | 5 | 10 | 20 | 40 | 80 |
> |--|-|-|-|-|-|-|-|
> | Gen PPL ↓ | 52.44 ± 1.83 | 47.17 ± 1.84 | 43.73 ± 1.43 | 42.35 ± 1.90 | 42.14 ± 1.39 | 43.57 ± 1.71 | 42.52 ± 1.98 |
>
> | MDLM (𝜎_attn) | 10 | 20 | 40 | 80 | 120 | 200 | 300 |
> |--|-|-|-|-|-|-|-|
> | Gen PPL ↓ | 11.10 ± 1.48 | 10.13 ± 0.81 | 10.02 ± 1.89 | 10.77 ± 0.98 | 11.05 ± 1.15 | 11.24 ± 0.31 | 11.37 ± 1.13 |
>
> Both models show broad stability for 𝜎_attn. Vanishing distance heuristic also works for this setting's tuning.

---

### Official Review · Reviewer_ESB4 · 2026-03-13

**Soundness:** 3
**Presentation:** 1
**Significance:** 3
**Originality:** 3
**Overall Recommendation:** 4
**Confidence:** 4

**Summary:**

The paper proposes SAKE, a training-free guidance method for discrete text diffusion models. It computes a diversity signal by maximizing order-2 Rényi entropy over a kernel Gram matrix that captures semantic similarity and positional information, then linearizes this in embedding space for efficiency. Results on code generation (HumanEval, MBPP) and math reasoning (GSM8K) show improvements, particularly at pass@k for large k.

**Compliance With Llm Reviewing Policy:**

Affirmed.

**Final Justification:**

I thank the authors for their response and appreciate the time they took to address my comments. Some of my questions were addressed, and the rebuttal clarified several points.

**Key Questions For Authors:**

* Can you show results on a second discrete diffusion model?

* What happens to pass@1 as γ increases? Is there a clean tradeoff curve?

**Strengths And Weaknesses:**

> Strengths:

The unified framework in Section 4.1 (Proposition 4.1) is clean and gives a nice lens to compare temperature scaling, D-CFG, and the proposed method under one roof. The observation that temperature scaling monotonically reduces entropy (Eq. 9) while SAKE can do both directions is well-motivated.


> Weaknesses:
* Single base model. All text experiments use LLaDA-8B. This is a fairly niche model. Without results on at least one other DLM (e.g., MDLM, or Dream), it's hard to know if the gains are specific to LLaDA's architecture/training or general to discrete diffusion. This is a significant gap for a methods paper.

* Pass@1 doesn't improve (or slightly degrades). Table 2 shows pass@1 going from 32.9 → 32.0 on HumanEval and 40.2 → 39.0 on MBPP. The paper frames everything around pass@k and self-consistency, but many practical settings care about single-sample quality.

* Hyperparameter sensitivity is absent. You have γ, σ, and σ_attn. The paper says "guidance strength optimized via grid search" but never shows sensitivity curves. How brittle is this? If the method requires careful per-task tuning of three hyperparameters, the "training-free" selling point is weaker than it appears.

* Limited baselines for the diversity claim. For text diversity, the only comparisons are temperature scaling and D-CFG. What about nucleus sampling, typical sampling, or min-p? These are standard and widely used. The related work mentions them but the experiments ignore them.

---

> ### Author Rebuttal · Authors · 2026-03-31
>
> We sincerely thank the reviewer for their constructive and insightful feedback. Below is our response to the comments and questions in the review.
>
> **Q1: Additional Base Model**
>
> Thanks for pointing this out. To address it, we tested base models MDLM and BD3LM in Q3. We generated samples using their official scripts and checkpoints, evaluating them with GPT-2-large generative perplexity (Gen PPL).
>
> Our results below show consistent gains across both models, proving our method generalizes beyond LLaDA's architecture. The revision will also include code and math benchmark results for reasoning models like Dream-7B.
>
> ---
> **Q2: Pass@1 Results**
>
> The reviewer notes pass@1 may not improve under diversity guidance. This is expected: increasing diversity shifts probability mass away from a single mode. However, our approach yields a better fidelity-diversity tradeoff than baselines.
>
> For pass@1, guidance strength γ can be lowered. To demonstrate, we quantified the γ–pass@1 trade-off on LLaDA HumanEval via a new γ sweep using non-deterministic sampling (T=0.2) across various seeds:
>
> | | γ = 0.0 (base) | γ = 0.05 | γ = 0.1 | γ = 0.2 | γ = 0.3 | γ = 0.5 | γ = 0.7 | γ = 1.0 |
> |-|-|-|-|-|-|-|-|-|
> | HumanEval pass@1 | 32.6 ± 3.5 | 32.5 ± 3.5 | 32.6 ± 3.5 | 32.0 ± 3.5 | 32.6 ± 3.5 | 30.5 ± 3.3 | 28.8 ± 3.2 | 21.1 ± 2.9 |
>
> Pass@1 remains stable for moderate γ (0.05–0.3). Stronger guidance (γ ≥ 0.5) reduces pass@1, as over-diversification degrades single-sample performance.
>
> ---
> **Q3: Hyperparameters**
>
> **Guidance strength γ**:
>
> Besides pass@1 results in Q2, we also present the γ–quality trade-off on MDLM and BD3LM.
>
> |Gen PPL ↓ | γ = 0.0 | γ = 0.05 | γ = 0.1 | γ = 0.2 | γ = 0.3 | γ = 0.5 | γ = 0.7 |
> |-|-|-|-|-|-|-|-|
> | MDLM | 18.47 ± 4.75 | 16.07 ± 5.38 | 13.85 ± 3.72 | 11.36 ± 1.72 | **10.13 ± 0.81** | 14.29 ± 4.68 | 22.23 ± 8.87 |
> | BD3LM | 60.88 ± 2.20 | 49.72 ± 1.25 | 47.34 ± 1.42 | **42.14 ± 1.39** | 56.17 ± 1.88 | 106.94 ± 3.62 | 253.90 ± 10.32 |
>
> With moderate γ (0.05–0.2), SAKE provides quality improvements over base model MDLM and BD3LM. These results are consistent with Humaneval results in Q1.
>
>
> **Semantic Kernel Bandwidth 𝜎**:
>
> 𝜎 sets the similarity detection threshold. Small 𝜎 values are highly sensitive to diversity (e.g., penalizing at synonyms level), while large 𝜎 enforce broader diversity. Crucially, 𝜎 controls the type of diversity, whereas guidance strength 𝛾 controls the amount.
>
> Gen PPL evaluations demonstrate that optimal 𝜎 values are stable across broad ranges:
>
> | MDLM (𝜎) | 5 | 10 | 20 | 30 | 40 | 80 | 120 |
> |--|-|-|-|-|-|-|-|
> | Gen PPL ↓ | 10.69 ± 1.69 | 10.43 ± 1.17 | 10.13 ± 0.81 | 10.78 ± 1.03 | 11.94 ± 1.02 | 16.53 ± 1.39 | 18.33 ± 2.08 |
>
> | BD3LM (𝜎) | 40 | 60 | 80 | 100 | 120 | 200 | 300
> |--|-|-|-|-|-|-|-|
> | Gen PPL ↓ | 43.69 ± 2.69 | 42.43 ± 2.17 | 42.14 ± 1.39 | 43.13 ± 2.31 | 44.39 ± 2.42 | 51.65 ± 3.65 | 57.11 ± 2.18 |
>
> MDLM is near-optimal for 𝜎 $\in$ [5, 30] and BD3LM for 𝜎 $\in$ [40, 120]. Thus, fine-grained tuning is unnecessary. Selecting 𝜎 within a broad reasonable range suffices for gains. We will include these sensitivity curves in the appendix.
>
> **Attention Kernel Bandwidth 𝜎_attn**:
>
> 𝜎_attn controls the guidance's positional receptive field. Larger 𝜎_attn allow influence from more distant tokens.
>
> We offer an interpretable initialization for 𝜎_attn. For a token distance $d$, define a threshold $\epsilon$ where attention vanishes:
>
> $$
> \exp\left(-\frac{d^2}{2\sigma_{\text{attn}}^2}\right) = \epsilon \implies d = \sigma_{\text{attn}} \sqrt{2 \ln\left(\frac{1}{\epsilon}\right)}
> $$
>
> If attention vanishes at 1% $(\epsilon=0.01)$, the vanishing distance is $d_v \approx 3.03 \sigma_{\text{attn}}$. Practitioners can set 𝜎_attn based on a desired vanishing distance (e.g. context length), minimizing tuning. Empirical sensitivity analysis shows stable optimal performance across a wide range of 𝜎_attn:
>
> | BD3LM (𝜎_attn) | 1 | 2 | 5 | 10 | 20 | 40 | 80 |
> |--|-|-|-|-|-|-|-|
> | Gen PPL ↓ | 52.44 ± 1.83 | 47.17 ± 1.84 | 43.73 ± 1.43 | 42.35 ± 1.90 | 42.14 ± 1.39 | 43.57 ± 1.71 | 42.52 ± 1.98 |
>
> | MDLM (𝜎_attn) | 10 | 20 | 40 | 80 | 120 | 200 | 300 |
> |--|-|-|-|-|-|-|-|
> | Gen PPL ↓ | 11.10 ± 1.48 | 10.13 ± 0.81 | 10.02 ± 1.89 | 10.77 ± 0.98 | 11.05 ± 1.15 | 11.24 ± 0.31 | 11.37 ± 1.13 |
>
> Both models show broad stability for 𝜎_attn. Vanishing distance heuristic also works for this setting's tuning.
>
> ---
>
> **Q4: Nucleus Sampling (top-p)**
>
> Thanks for raising this. To address your question on Nucleus sampling, we swept top-p values for the base LLaDA model on HumanEval:
>
> | LLaDA (top-p) | 0.5 | 0.6 | 0.7 | 0.8 | 0.9 | 0.95 | 1.0 |
> |--|-|-|-|-|-|-|-|
> | HumanEval pass@32 | 35.4 | 36.6 | 40.9 | 40.9 | 40.2 | **41.2** | 39.6 |
>
> These results confirm that the standard HumanEval top-p=0.95 default yields the best pass@32 performance, validating it as the optimal choice. We will include this discussion on Nucleus sampling and its other results in the revision.

---

> > ### Author Rebuttal · Reviewer_ESB4 · 2026-04-04
> >
> > I thank the authors for their response and appreciate the time they took to address my comments. Some of my questions were addressed, and the rebuttal clarified several points.

---

> > > ### Author Response · Authors · 2026-04-05
> > >
> > > We sincerely thank Reviewer ESB4 for the constructive review and positive feedback on our response. We are glad that our rebuttal could clarify several of the raised points and address the comments.

---

### Decision · Program_Chairs · 2026-04-30

**Decision:**

Accept (regular)

**Comment:**

The reviewers unanimously agree that the proposed Semantic-Aware Kernel Entropy (SAKE) guidance is a technically sound and intuitive approach to improving the fidelity-diversity trade-off in discrete text diffusion models.

Initial concerns regarding the reliance on a single base model, hyperparameter sensitivity, and potential inference latency were comprehensively addressed during the rebuttal. The authors provided convincing additional experiments on MDLM and BD3LM, along with detailed stability and throughput analyses that successfully resolved the reviewers' reservations.

Given the solid theoretical formulation, the computational efficiency of the embedding-space linearization, and the clear empirical improvements on reasoning-intensive tasks, this paper provides a valuable contribution to the generative modeling community. I recommend acceptance.